# Diverging responses of high latitude $CO_2$ and $CH_4$ emissions in idealized climate change scenarios

Philipp de Vrese[1], Tobias Stacke[2], Thomas Kleinen[1], and Victor Brovkin[1,3]

[1]Max Planck Institute for Meteorology, The Land in the Earth System, Hamburg, 20146, Germany
[2]Helmholtz-Zentrum Geesthacht, Institute of Coastal Research, Geesthacht, 21502, Germany
[3]University of Hamburg, Center for Earth System Research and Sustainability, Hamburg, 20146, Germany

**Correspondence:** Philipp de Vrese (philipp.de-vrese@mpimet.mpg.de)

**Abstract.** The present study investigates the response of the high latitude carbon cycle to changes in atmospheric greenhouse gas (GHG) concentrations in idealized climate change scenarios. To this end we use an adapted version of JSBACH – the land surface component of the Max Planck Institute for Meteorology Earth System Model (MPI-ESM) – that accounts for the organic matter stored in the permafrost-affected soils of the high northern latitudes. The model is run under different
climate scenarios that assume an increase in GHG concentrations, based on the Shared Socioeconomic Pathway 5 and the Representative Concentration Pathway 8.5, which peaks in the years 2025, 2050, 2075 or 2100, respectively. The peaks are followed by a decrease in atmospheric GHGs that returns the concentrations to the levels at the beginning of the 21st century, reversing the imposed climate change. We show that the soil $CO_2$ emissions exhibit an almost linear dependence on the global mean surface temperatures that are simulated for the different climate scenarios. Here, each degree of warming increases the
fluxes by, very roughly, 50% of their initial value, while each degree of cooling decreases them correspondingly. However, the linear dependence does not mean that the processes governing the soil $CO_2$ emissions are fully reversible on short timescales, but rather that two strongly hysteretic factors offset each other – namely the net primary productivity and the availability of formerly frozen soil organic matter. In contrast, the soil methane emissions show a less pronounced increase with rising temperatures and they are consistently lower after the peak in the GHG concentrations than prior to it. Here, the net fluxes
could even become negative and we find that methane emissions will play only a minor role in the northern high latitude contribution to global warming, even when considering the high global warming potential of the gas. Finally, we find that at a global mean temperature of roughly 1.75 K ($\pm$ 0.5 K) above pre-industrial levels the high latitude ecosystem turns from a $CO_2$ sink into a source of atmospheric carbon, with the net fluxes into the atmosphere increasing substantially with rising atmospheric GHG concentrations. This is very different from scenario simulations with the standard version of the MPI-ESM
in which the region continues to take up atmospheric $CO_2$ throughout the entire 21st century, confirming that the omission of permafrost-related processes and the organic matter stored in the frozen soils leads to a fundamental misrepresentation of the carbon dynamics in the Arctic.

*Copyright statement.* TEXT

# 1 Introduction

High latitude terrestrial ecosystems are increasingly recognised as an important factor for the global carbon cycle. On the one hand, global warming is expected to increase the vegetation cover and primary productivity – a trend termed Arctic greening (Keenan and Riley, 2018; Pearson et al., 2013; Zhang et al., 2018), which could significantly increase the terrestrial uptake of atmospheric $CO_2$ (Qian et al., 2010; McGuire et al., 2018). On the other hand, there are large quantities of effectively inert organic matter stored within the frozen soils of the Northern Hemisphere and a significant fraction of these could become exposed to microbial decomposition in a warmer climate. Areas underlain by permafrost, defined by soil temperatures below the freezing point for at least 2 consecutive years, contain 1100 - 1700 Gt of carbon, the largest fraction of which is stored within the frozen part of the ground (Zimov, 2006a; Tarnocai et al., 2009; Hugelius et al., 2014). With the temperature increase in the high latitudes being about twice as large as the global average (Stocker et al., 2013), the last decades have already seen substantial changes in the permafrost-affected regions. Regional soil temperatures have increased by up to 2 K and there is a pronounced reduction in the extent of permafrost-affected areas combined with an increase in active layer depth, which leaves large quantities of organic matter vulnerable to decomposition (Biskaborn et al., 2019; Stocker et al., 2013; Etzelmueller et al., 2011; Osterkamp, 2007; Shiklomanov et al., 2010; Frauenfeld, 2004; Wu and Zhang, 2010; Callaghan et al., 2010; Isaksen et al., 2007; Brown and Romanovsky, 2008; Romanovsky et al., 2010).

Depending on the assumed greenhouse gas (GHG) emissions, climate change scenarios project the Arctic temperatures to increase by between 3 K and 8 K until the end of the 21st century (Stocker et al., 2013). Many modelling studies have investigated the resulting decrease in organic matter stored in the permafrost-affected regions and for the high emission scenarios – corresponding to a temperature increase of 8 K – the soils are expected to emit around $120 \pm 80$ Gt of carbon until the year 2100 (Schuur et al., 2013; Schaefer et al., 2014; McGuire et al., 2018). Increasing temperatures also accelerate the Arctic greening trend and it is highly uncertain at which point the carbon release from thawing soils would surpass the additional carbon uptake by vegetation. However, it is generally assumed that the Arctic ecosystem will turn from a carbon sink into a carbon source within the 21st century (Schuur et al., 2008; Schaefer et al., 2011; Koven et al., 2015; Schuur et al., 2015; McGuire et al., 2018; Parazoo et al., 2018; Natali et al., 2019). The (net) carbon release will further increase the atmospheric GHG concentrations, leading to a positive feedback. Studies indicate that this feedback will not only notably accelerate the global warming for high emission scenarios, which result in a near-disappearance of the terrestrial near-surface permafrost (often defined as being located within the top 3 m of the soil), but even for the temperature target of the Paris Agreement (MacDougall et al., 2012; Burke et al., 2017b, 2018; Comyn-Platt et al., 2018).

It is exceedingly difficult to estimate the Arctic contribution to future warming. One issue is the timescale on which the carbon would be released from permafrost-affected soils. While local observations indicate that the processes which affect the soil carbon emissions are often locally confined and act on very short timescales, large-scale models do not represent these small-scale processes. Thus, studies relying on these models suggest that the increase in emissions is likely to occur gradually

over a timescale of hundreds of years (Schuur et al., 2015). Another important issue is the fraction of carbon that is released in the form of $CH_4$ rather than $CO_2$. Methane is a much more potent GHG (Stocker et al., 2013) and even a small fraction of formerly frozen carbon that is released as $CH_4$ would increase the respective global warming potential substantially. Methane is produced during the decomposition under anaerobic conditions, requiring soils to be water-saturated. Hence, future methane emissions are highly dependent on changes in the sub-surface hydrology in permafrost-affected regions (Olefeldt et al., 2012). It is difficult to represent saturated soils at the typical spatial resolution of present-day Earth system models, making it hard to determine the areas in which the decomposition occurs under anaerobic conditions. Furthermore, the hydrological response to permafrost degradation is very complex and there is some disagreement between land-surface models even as to whether high latitude soils would in general become drier or wetter in the future (Berg et al., 2017; Andresen et al., 2020). Thus, there are comparatively few studies that use large-scale models to investigate the change in soil methane emissions for future warming scenarios (Lawrence et al., 2015; Burke et al., 2012; von Deimling et al., 2012; Koven et al., 2015; Oh et al., 2020).

The present study aims at improving our understanding of the importance of the Arctic ecosystem for the global carbon cycle not only by presenting additional estimates of the carbon fluxes under a future warming scenario. More importantly, the goal of the study is to provide a better understanding of the processes that govern these fluxes – in particular the soil methane emissions – in permafrost-affected regions. The current anthropogenic GHG emissions make it increasingly likely that temperatures will overshoot any temperature target before atmospheric GHG concentrations could be stabilized at a desirable level (Geden and Löschel, 2017; Parry et al., 2009; Huntingford and Lowe, 2007; Nusbaumer and Matsumoto, 2008; Ricke et al., 2017; Rogelj et al., 2015, 2018). But while many studies have investigated the response of the Arctic ecosystem to increasing GHG concentrations, only few studies exist that investigate its response to a decrease in concentrations (Boucher et al., 2012; Eliseev et al., 2014) and it is still an open question how the high latitude carbon cycle responds to overshooting temperatures. Thus, we do not only target the response of the system to increasing temperatures, but also during a subsequent temperature decline.

Our investigation is based on simulations with the land surface component of the MPI-ESM1.2 (Mauritsen et al., 2019), the latest release of the Max Planck Institute for Meteorology Earth System Model. However, the standard JSBACH model includes a number of parametrizations that are not well suited for the specific conditions that are characteristic of the high latitudes, e.g., the standard model does not account for freezing and melting of soil water and estimates the decomposition rates of soil organic matter based on the conditions at the surface. As a result, the standard model has certain shortcomings in the representation of the high latitude energy-, water- and carbon cycle, such as a strong overestimation of the thaw-depths and an inability to conserve the effectively inert organic matter contained in the permafrost (Fig. 1). In the following we will describe the required modifications to the model, together with a more detailed description of the simulations that were performed in the context of this study (Sec. 2). Section 3 details our findings with respect to the soil $CO_2$ and $CH_4$ emissions under in- and decreasing temperatures, while section 4 discusses them in the context of the global carbon cycle.

## 2 Methods

### 2.1 Model

The changes that were made to JSBACH include the implementation of 3 new modules that represent the formation of inundated areas and wetlands (both described in sec. 2.1.3) as well as the soil methane production including the gas transport in soils (sec. 2.1.4). Furthermore, we adapted the representation of the soil physics and the carbon cycle to include processes that are relevant for permafrost-affected regions.

#### 2.1.1 Soil carbon

In JSBACH, the soil carbon dynamics are simulated by the YASSO model, which calculates the decomposition of organic matter at and below the surface considering five different lability classes – acid-hydrolyzable, water-soluble, ethanol-soluble, non-soluble/non-hydrolyzable and a more recalcitrant humus pool (Liski et al., 2005; Tuomi et al., 2011; Goll et al., 2015). The decomposition rates are determined by the standard mass loss parameter, which differs between the lability classes, and two factors that account for the temperature and moisture dependencies of the decomposition process. The standard YASSO model does not consider a vertical distribution of the organic matter within the soil and the decomposition rates depend on the simulated surface temperatures and precipitation rates. This approach works well in regions in which most of the soil carbon is stored close to the surface, but it is problematic for permafrost-affected regions. The vertical carbon transport in these regions is dominated by very effective processes – cryoturbation (Schuur et al., 2008) – and soils can store organic matter in depth of several meters. Thus, the conditions under which this organic matter decomposes are not well approximated by surface temperatures and precipitation rates. To improve the representation of the carbon cycle in permafrost-affected regions, we implemented a vertical structure of the soil carbon pools and calculate the decomposition rates using depth-dependent soil temperature and liquid soil water content. Furthermore, we added a simple parametrization to distribute the carbon inputs according to idealized root profiles and a scheme to account for the accumulation of organic matter at the top of the soil column and the vertical transport due to bio- and cryoturbation.

**Structure of the soil carbon pools**

JSBACH distinguishes between above and below ground carbon pools. In the standard model this separation is only relevant for the computation of the fuel load required by the fire module. However, fresh litter at the surface, such as branches or leaves, has very different thermophysical and hydrological properties than organic matter that is encompassed in the soil. To be able to account for these differences, the new structure maintains the separation of above and below ground carbon but introduces a vertical discretization of the below ground carbon pools. As we also maintain the conceptual structure of the lability classes, the new scheme represents soil carbon by 5 lability classes on every model soil layer and 4 above ground carbon pools (note that the humus pool does not exist at the surface, see below).

The present model version distinguishes between anoxic and oxic decomposition in the inundated and the non-inundated fractions of the grid box (see below) and the soil carbon pools need to be separated accordingly. Here, we do not simulate the respective pools explicitly. Instead we determine $r_{C_{in}}^{t_{end}}$, the ratio between the carbon concentrations in the inundated ($C_{in}^{t_{end}}$) and the non-inundated ($C_{dry}^{t_{end}}$) fractions, for each of the soil carbon pools after the decomposition is computed in timestep $t$.

$$r_{C_{in}}^{t_{end}} = \frac{C_{in}^{t_{end}}}{C_{dry}^{t_{end}}} \tag{1}$$

In the consecutive time step $t+1$, the soil carbon is distributed between inundated and non-inundated carbon pools according to $r_{C_{in}}^{t_{end}}$ before the decomposition is calculated.

$$C_{in}^{t+1_{start}} = C_{tot}^{t+1_{start}} \left(1 + \frac{1}{r_{C_{in}}^{t_{end}}}\right)^{-1}, \tag{2}$$

$$C_{dry}^{t+1_{start}} = C_{tot}^{t+1_{start}} - C_{in}^{t+1_{start}}. \tag{3}$$

For changes in the inundated area, $r_{C_{in}}$ is updated between two calls of the decomposition routine. This approach allows us to separate oxic and anoxic respiration without having to calculate the entirety of relevant processes – such as land cover changes, disturbances, etc. – for two sets of carbon pools.

**Carbon inputs**

In JSBACH, the litter inputs are divided into above and below ground litter fluxes, with 70% of the coarse and 50% of the fine litter entering the above ground pools. We maintain this separation but distribute the below ground litter inputs among the vertical soil layers according to vegetation type specific root profiles. Similarly, the below ground carbon inputs that result from disturbances and land use change as well as root exudates are distributed according to these profiles. The cumulative root fraction, $Y$, is described by:

$$Y = 1.0 - \beta^z, \tag{4}$$

with $z$ being the depth below the surface [$cm$]. The parameter $\beta$ is taken from Jackson et al. (1996) and matched to the plant functional types employed by JSBACH. Furthermore, the cumulative root fraction is scaled to a maximum depth, which is limited by the lower of either the prescribed rooting depth or the previous year maximum thaw depth. The latter is done because JSBACH uses a rooting depth that is fixed in each grid box, but we assume that plants do not extend their roots into the perennially frozen regions of the soil.

**Transport**

The vertical carbon transport in permafrost-affected regions is dominated by frost heave and freeze-thaw cycles (Schuur et al., 2008). However, cryoturbation involves a variety of complex processes that depend on small-scale features of the soil and even

though process models exists (Peterson et al., 2003; Nicolsky et al., 2008), these are not applicable on the scales of land surface models. Thus, we follow the approach of Koven et al. (2009, 2013) and describe the vertical mixing of soil organic matter as a diffusive transport:

$$\frac{\partial C_{lc,z}}{\partial t} = \frac{\partial}{\partial z}\left(D(z)\frac{\partial C_{lc}}{\partial z}\right),$$ (5)

with $C$ being the carbon concentration of the lability class $lc$, $D$ the diffusion coefficient and $z$ the depth below the surface.

Similar to Burke et al. (2017a) we use a constant diffusivity – not varying between grid boxes – to represent bioturbation in regions that are not affected by near-surface permafrost. At the surface we use a diffusivity of 1.5 cm$^2$ year$^{-1}$ and for the deeper layers we assume the mixing rates to decline linearly with increasing depth up to a maximum depth of 3 meters or up

to the bedrock depth. In permafrost regions the mixing rates are much larger and vary based on soil conditions. It is assumed that cryoturbation is more effective in wetter soils and when freezing during winter and thawing during spring extends over a long periods – weeks to month – during which the soil repeatedly thaws and refreezes. To account for these effects, we assume a maximum diffusivity of 15 cm$^2$ year$^{-1}$ which is scaled by two terms representing the (previous year mean) saturation of the active layer and the number of days in which temperatures crossed the freezing point. At the surface, diffusivity $D$ [cm$^2$

165   year$^{-1}$] is given by:

$$D(s) = \begin{cases} 1.5, & \text{for bioturbation} \\ 15 \cdot w_{atl} \cdot min\left(1, \frac{N_{dc0}}{N_{dc0,ref}}\right), & \text{for cryoturbation} \end{cases}$$ (6)

where $w_{atl}$ is the saturation of the active layer, $N_{dc0}$ the number of days per year in which surface temperatures crossed the freezing point and $N_{dc0,ref}$ a respective reference value which was set to 40 days year$^{-1}$. For the depth dependence of the mixing rates in permafrost-affected regions there are two options included in the scheme. Either a constant diffusivity is

assumed throughout the active layer (or until the border with the bedrock), or the mixing rates are assumed to decline linearly throughout the active layer.

The present model structure separates the organic matter into above and below ground pools and the vertical mixing described above is only applied to the below ground carbon. The organic matter that is deposited above the surface needs to be

incorporated into the soil before it can be transported into the deeper layers. The separation between above and below ground litter is a mere conceptual one, used to account for the different properties of the organic matter, and the above ground litter occupies the same physical space as the below ground pools representing the top soil layer. Hence, the transfer of carbon from the above to the below ground pools requires a change in properties rather than in space, and there are two ways by which this can happen. The decomposition at the surface turns a given fraction of the organic matter into humus and with this transfor-

mation we assume a change in physical properties that reassigns the carbon from the above to the below ground pools (hence there is no above ground humus pool). Furthermore, organic matter builds up on the surface in grid boxes in which the long term carbon input at the surface is larger than the respiration rates. Here, we assume the load of organic matter on the surface

to affect its properties, as the latter are largely dependent on the bulk density which is reduced under pressure. Thus, the excess material is transferred to the corresponding below ground pools when the load of organic matter exceeds a given threshold – for the present study we choose $\approx 10\,\mathrm{kg\,m^{-2}}$. Assuming a litter density of $\approx 75\,\mathrm{kg\,m^{-3}}$, this corresponds to a surface organic layer with a maximum depth of around 15 cm – averaged over the grid box area – which is well within the range of typical organic layer thickness (Yi et al., 2009; Lawrence et al., 2008; Johnstone et al., 2010) and very similar to the soil organic layer used in the study of Ekici et al. (2014).

**Decomposition**

With respect to the decomposition rates, $k_{lc}$, we follow the same approach as the standard YASSO model in which a mass loss parameter $\alpha_{lc}$, specific to the lability class, is multiplied by factors accounting for the temperature and moisture dependencies of decomposition – $d_{temp}$ and $d_{mois}$:

$$k_{lc} = \alpha_{lc} \cdot d_{temp} \cdot d_{mois} \tag{7}$$

For the above ground carbon pools we use the parametrizations of the standard model:

$$d_{temp}(s) = \exp(\beta_1 \cdot T_{surf} + \beta_2 \cdot T_{surf}^2) \qquad\qquad ; \text{with } \beta_1 = 0.095 \text{ and } \beta_2 = -0.0014, \tag{8}$$

$$d_{mois}(s) = 1 - \exp(\gamma \cdot P) \qquad\qquad ; \text{with } \gamma = -1.21, \tag{9}$$

where $T_{surf}$ is the surface temperature $[^{\circ}C]$ and $P$ the precipitation rate $[m\ year^{-1}]$. To account for the different decomposition rates under aerobic and anaerobic conditions we calculate the moisture dependence in inundated areas as Kleinen et al. (2019):

$$d_{mois,inu}(s) = d_{mois}(s) \cdot 0.35 \tag{10}$$

It should be noted that we assume that only a fraction of the above ground organic matter in inundated areas decomposes under anaerobic conditions. As discussed above, the above ground carbon pools in the model occupy the same physical space as the below ground pools representing the top layer of the soil column. In reality, however the litter that falls on top of a fully saturated soil column would still decompose aerobically unless there is standing water on top of the surface. Even then it is highly uncertain how much of the litter decomposes under anaerobic or aerobic conditions as this depends very much on the shape of the litter and on the depth of the standing water – a twisted branch may be located largely above the water while a straight branch would be fully submerged. In the model we deal with this uncertainty by including the fraction of the above ground organic matter that decomposes anaerobically as an input parameter that can be varied between simulations (see below).

For the below ground decomposition rates, we evaluated a variety of functions to represent the moisture and temperature dependencies (Sierra et al., 2015), some of which are included as options in the present version of JSBACH. The goal of this evaluation was to establish a combination of dependencies that changes the carbon dynamics in the non-permafrost-affected regions as little as possible, while preserving the organic matter stored within the perennially frozen ground. For this study,

we chose the temperature dependence parametrization of the YASSO model in combination with a simplified version of the moisture limitation function used in the CENTURY ecosystem model (Kelly et al., 2000). The temperature and moisture dependencies, $d_{temp}(z)$, $d_{mois}(z)$ and $d_{mois,inu}(z)$ in depth $z$ ($z \neq s$) are given by:

$$d_{temp}(z \neq s) = \exp(\beta_1 \cdot T_z + \beta_2 \cdot T_z^2), \tag{11}$$

$$d_{mois}(z \neq s) = 1.2 \cdot \left(\frac{w_z^* - b}{a - b}\right)^{d \cdot \frac{b-a}{a}} \cdot \left(\frac{w_z^*}{a}\right)^d \qquad ; \text{with } a = 0.575, \, b = 1.5 \text{ and } d = 3, \tag{12}$$

$$d_{mois,inu}(z \neq s) = 1.2 \cdot \left(\frac{1 - b}{a - b}\right)^{d \cdot \frac{b-a}{a}} \cdot \left(\frac{1}{a}\right)^d \qquad ; \text{for } w_{liq,z} > w_{ice,z}, \tag{13}$$

where $T_z$ is the temperature in depth $z$ and $w_z^*$ represents the relative saturation of the soil, considering only the liquid water content. Note however, that we do not use the saturation of the soil directly, because the formulation of the soil hydrology module prevents the soil moisture from dropping below a certain threshold or to increase above the field capacity. In order to account for this, $w_z^*$ is not given relative to the soils pore space, but relative to the range between the wilting point and the field capacity. In addition, we apply a subgrid-scale distribution of the soil water in order to determine the inundated grid box fraction (see below). Thus $w_z^*$ does not correspond to the mean saturation of the grid box but to the saturation of the non-inundated fraction. In the inundated fraction soils are fully saturated and $d_{mois,inu}(z \neq s)$ has a fixed value of 0.32. It is assumed, however, that decomposition in the inundated areas can only occur when the liquid water content in a soil layer ($w_{liq,z}$) is larger than the ice content ($w_{ice,z}$), even though it should be noted that microbes in reality do not necessarily require free water in the soil to survive and can maintain viability for thousands of years within frozen soils (Gilichinsky et al., 2003). However, we assume negligible activity under these conditions.

### 2.1.2 Permafrost physics

The representation of the physical processes in the soil that are related to permafrost are largely based on the implementation of Ekici et al. (2014). However, there are certain important differences, which will be described in more detail in the following. Most importantly, we adapted the representation of soil organic matter from a pervasive organic top soil layer to explicitly simulating the organic matter at the surface and within each of the vertical soil layers. Furthermore, we adapted the formulations of transpiration and the water limitations of plants to account for perennially frozen soils. Finally, the model accounts for the heat generated by decomposition (Khvorostyanov et al., 2008b), even though the effects are negligible in all the simulations.

### Soil properties

The present model version represents the organic matter above and below the surface explicitly and accounts for the respective effects on a given soil property, $X_{soil}(z)$, by aggregating the respective properties of organic, $X_{org}(z)$, and mineral material, $X_{min}$, according to their volumetric fractions, $f_{org}(z)$ and $(1 - f_{org}(z))$:

$$X_{soil}(z) = f_{org}(z) \cdot X_{org}(z) + (1 - f_{org}(z)) \cdot X_{min}. \tag{14}$$

The fraction of organic matter, $f_{org}(z)$, is given by:

$$f_{org}(z) = \frac{\rho_c(z)/r_{c2b}}{\rho_{org}(z)}, \tag{15}$$

where $\rho_c(z)$ is the mass concentration of carbon at depth $z$, $r_{c2b}$ the carbon to biomass ratio and $\rho_{org}(z)$ the dry bulk density of organic matter. The estimates of $\rho_{org}(z)$ vary strongly depending on the quality of organic matter and whether it pertains to litter at the surface or to organic matter that is integrated in the soil (O'Donnell et al., 2009; Ahn et al., 2009; Chojnacky et al., 2009). For the present study we chose $\rho_{org}(s) = 75$ kg m$^{-3}$ for above ground organic matter and $\rho_{org}(z \neq s) = 150$ kg m$^{-3}$ for the organic matter below ground. Likewise the properties of the organic matter, $X_{org}(z)$, differ between above and below ground organic matter (Peters-Lidard et al., 1998; Beringer et al., 2001; O'Donnell et al., 2009; Ahn et al., 2009; Chojnacky et al., 2009; Ekici et al., 2014). $r_{c2b}$ was set to 0.5.

This aggregation was applied to all soil properties with the exception of the saturated hydraulic conductivity, for which we follow the approach of the Community Land Model (Oleson et al., 2013). Here, it is assumed that connected flow pathways form, once the fraction of organic matter exceeds a certain threshold. These need to be accounted for in the bulk hydraulic conductivity, $k_{sat}(z)$:

$$k_{sat}(z) = f_{uncon}(z) \cdot k_{sat,uncon}(z) + (1 - f_{uncon}(z)) \cdot k_{sat,org}(z) \tag{16}$$

where $f_{uncon}(z)$ is the grid box fraction in which no conected pathways exist, $k_{sat,uncon}(z)$ the saturated hydraulic conductivity in this fraction and $k_{sat,org}(z)$ the conductivity in the grid box fraction in which pathways form.

$$k_{sat,uncon}(z) = f_{uncon}(z) \cdot \left( \frac{1 - f_{org}(z)}{k_{sat,min}} + \frac{f_{org}(z) - f_{perc}(z)}{k_{sat,org}(z)} \right)^{-1} \qquad \text{; with} \tag{17}$$

$$f_{uncon}(z) = 1 - f_{perc(z)} \qquad \text{; and} \tag{18}$$

$$f_{perc}(z) = (1 - f_{thresh})^{-\beta_{perc}} \cdot (f_{org}(z) - f_{thresh})^{\beta_{perc}} \qquad \text{; for } f_{org}(z) \geq f_{thresh}, \text{ and} \tag{19}$$

$$f_{perc}(z) = 0 \qquad \text{; for } f_{org}(z) < f_{thresh}, \tag{20}$$

where $\beta_{perc} = 0.139$ and $f_{thresh} = 0.5$.

**Soil and surface hydrology**

A given fraction of the water within the soil remains liquid even at sub-zero temperatures. In reality, supercooled water exists in the presence of certain chemicals, such as salts, that lower the freezing temperature, but also because of the absorptive and capillary forces that soil particles exert on the surrounding water. The model does not represent the soil chemical composition and we only account for the thin film of supercooled water that forms around the soil particles, which can be described by a freezing-point depression (Ekici et al., 2014; Niu and Yang, 2006). However, the liquid water is bound to the soil particles and it is questionable whether it is able to move through the surrounding soil-ice matrix. Thus, we assume the supercooled liquid water in the soil to be immobile in the present model version. As the vertical movement of water requires flow pathways to

be available, percolation of liquid water within the soil is inhibited when more than half of the soil pore space is occupied by ice.

Additionally, the standard model version assumes lateral drainage from all soil layers located above the bedrock. This drainage component is included to account for vertical channels, e.g., connected pathways in coarse material, cracks or crevices, that are assumed to be present in the large, heterogeneous grid cells at the standard resolution ($1.9° \times 1.9°$). These efficiently transport the water deeper underground towards the border between soil and bedrock where it runs of as base flow. However, in the presence of permafrost, we assume these vertical channels to be predominantly blocked by ice and we allow lateral drainage

only at the bedrock boundary or from those layers below which the soil is fully water saturated, i.e. at field capacity. These limitations on lateral drainage in combination with the inhibition of percolation for large ice contents facilitate high moisture levels within the active layer and the formation of a perched highly saturated zone on top of the perennially frozen soil layers, which are typical for permafrost regions (Swenson et al., 2012). Finally, we changed the conditions controlling infiltration at the surface. In the standard model, infiltration is partly temperature dependent, with no infiltration below the melting point.

This condition was removed so that infiltration is controlled purely by the saturation of the near-surface soil and the topography within the grid cell.

In JSBACH, transpiration and water stress are calculated based on the degree of saturation within the root zone. However, the respective parametrizations become very problematic in the presence of soil ice because they use a fixed parameter, the

maximum root zone soil moisture, relative to which the degree of saturation is calculated. In reality, the root zone in permafrost-affected regions is confined to depths above the perennially frozen regions of the soil, while the root zone can not adapt to the permafrost table in the standard model. Thus, the parametrization can result in plants experiencing constant water stress when the permafrost extends into the root zone, even if there is sufficient liquid water available in the upper layers. Similarly, bare soil evaporation is determined by the saturation of the top 6.5 cm of the soil, considering only the liquid water content relative

to the entire pore space not to the ice free pore space. Consequently, evaporation can be reduced substantially when there is ice in the top soil layer, despite enough liquid water being present at the surface. In the present model version we deal with this issue by accounting for the presence of ice and computing the saturation of the root zone and the top soil layer relative to the ice-free pore space.

### 2.1.3   Wetlands and inundated areas

In its standard version, JSBACH accounts neither for surface water bodies nor for inundated areas. For the present study we implemented two schemes that represent different aspects of their formation. The first scheme simulates the effect of ponding – the formation of wetlands because water can not infiltrate fast enough and pools at the surface, while the second scheme accounts for inundated areas that form in highly saturated soils due to low drainage fluxes. Note that in the result section we make no differentiation between wetlands and inundated areas because they have a very similar effect on the carbon cycle, in

that they both constitute areas in which soil organic matter decomposes under anaerobic conditions.

The ARNO model, which is used by JSBACH to determine the infiltration rates, does not account for ponding effects. Instead all water arriving on the soil surface is either infiltrated or converted into surface runoff (Dümenil and Todini, 1992; Todini, 1996). In the present version of JSBACH, we implemented a WEtland Extent Dynamics (WEED) scheme based on a concept developed for the global hydrology model MPI-HM (Stacke and Hagemann, 2012). WEED adds a water storage to the land surface which intercepts rainfall and snow melt prior to soil infiltration and runoff generation. Based on the surface area fraction $f_{pond}$ and the depth $h_{pond}$ of the storage, evaporation $E_{pond}$ and outflow $R_{pond}$ are computed as

$$E_{pond} = (1 - f_{snow}) \cdot (f_{pond} - f_{skin}) \cdot E_{pot} \tag{21}$$

$$R_{pond} = h_{pond} \cdot \frac{1}{(1 - f_{pond}) \cdot \lambda_{pond}} \tag{22}$$

Outflow accounts for topography in form of the outflow lag $\lambda_{pond}$ computed based on the orographic standard deviation $\sigma_{\text{oro}}$:

$$\lambda_{pond} = \left( 1 - \left( \frac{\sigma_{oro}}{\sigma_{max}} \right)^{\frac{1}{4}} \right) \cdot \lambda_{max} \tag{23}$$

resulting in an increased outflow when either the storage contains a large amount of water or the orographic variability in the grid cell is high. Runoff is subdivided into direct infiltration and lateral runoff. The former is diagnosed as the soil moisture saturation deficit of the uppermost soil layer for the wetland-covered grid cell fraction and added to the soil moisture storage directly. The latter is further processed into surface runoff and soil infiltration according to the standard soil scheme (Hagemann and Stacke, 2015; Dümenil and Todini, 1992). Runoff is assumed to be zero when temperatures fall below the freezing point. Considering all these fluxes, the water storage $S_{pond}$ changes according to:

$$\Delta S_{pond} = P_{rain} + P_{melt} - E_{pond} - R_{pond} \tag{24}$$

Due to the coarse model resolution it is not reasonable to quantify $f_{pond}$ for a given storage state explicitly from highly resolved topographical data. Instead, we attribute any change in the water volume of the wetland $V_{pond} = S_{pond} \cdot f_{pond} \cdot A_{cell}$ to changes in the depth and extent of the wetland using the topographical standard deviation of the grid cell:

$$\Delta h_{pond} = \left( \Delta V_{pond} \cdot \frac{\sigma_{oro}}{\sigma_{crit}} \right)^{\frac{1}{3}} \tag{25}$$

$$\Delta A_{pond} = \frac{V_{pond}}{h_{pond} + \Delta h_{pond}} - A_{pond} \tag{26}$$

Thus, any change in surface water is divided equally between water depth and extent if the orographic standard deviation of the grid cell equals a given critical orography standard deviation $\sigma_{crit}$. Thus, cells with a high orographic variation exhibit rather deep but small inundated fractions, while flat cells result in very shallow but extensive inundated fractions with a strong seasonality.

The WEED scheme is able to represent a realistic wetland distribution with extensive wetlands in the high northern latitudes and tropical rainforest regions. However, an extensive evaluation of the simulated water bodies is beyond the scope of the current study.

To determine the extent of inundation areas dynamically, we use an approach based on the TOPMODEL hydrological framework (Beven and Kirkby, 1979). TOPMODEL employs sub-grid-scale topographic information contained in the compound topographic index (CTI) to redistribute the grid-cell mean water table, raising the sub-grid-scale water table in areas of high CTI and lowering it where CTI is low. We employ the CTI index product by Marthews et al. (2015) for the CTI index at a resolution of 15 arcseconds to determine the distribution of CTI values within any particular grid cell and thus determine the fraction of the grid cell where the water table is at or above the surface. A detailed description of the approach is given by Kleinen et al. (2019).

### 2.1.4 Gases in the soil

The standard version of JSBACH does not differentiate between aerobic and anaerobic soil respiration. In order to determine the methane emissions from saturated soils, we implemented the methane model proposed by Kleinen et al. (2019). Based on Riley et al. (2011), the model determines $CO_2$ and $CH_4$ production in the soil, the transport of $CO_2$, $CH_4$ and $O_2$ through the three pathways diffusion, ebullition and plant aerenchyma, as well as the oxidation of methane wherever sufficient oxygen is present. Partitioning of the anaerobic decomposition product ($R_{anox}$) into $CO_2$ and $CH_4$ ($P_{CH_4}$) is temperature-dependent, with a baseline fraction of $CH_4$ production $f_{CH_4} = 0.4$ and a Q10 factor for $f_{CH_4}$ of Q10 = 1.5, with a reference temperature ($T_{ref}$) of 295K.

$$P_{CH_4} = R_{anox} \cdot f_{CH_4} \cdot Q10^{\frac{T_z - T_{ref}}{10}} \tag{27}$$

In each grid cell the methane model determines $CH_4$ production and transport for two grid cell fractions, the aerobic (non-inundated) and the anaerobic (inundated) fraction of the grid cell. If the inundated fraction changes, the amounts of $CO_2$, $CH_4$ and $O_2$ are conserved, transferring gases from the shrinking fraction to the growing fraction, proportional to the area change. Thus the model captures not just the emission of methane from inundated areas, but also the uptake and oxidation of methane by the soil in the non-inundated areas. It should be noted that the model also simulates the $CH_4$ emissions from wildfires and termites. However, we neglect these fluxes in the detailed discussion of the methane emissions and exclusively report the fluxes from wetlands and inundated areas as the focus of this study is on soil emissions.

## 2.2 Experimental setup

### 2.2.1 Simulations

The modifications described above change the behaviour of the model substantially, which introduces large additional uncertainties. These involve uncertainties that originate from the parametrizations themselves, but also from their interactions with other processes in the model. To account for these uncertainties we created an ensemble of 20 simulations in which key parameter values and parametrizations were varied (see below). However, the ensemble size, in combination with the temporal extent of the simulations, made it infeasible to use the fully coupled MPI-ESM which has roughly a hundred times the computational

demand of the land surface model. Instead we use JSBACH in an offline-setup, in which the land surface model is driven
by output from the fully coupled model. Here, we use output from simulations (10th ensemble member) with the standard
version of the MPI-ESM1.2 that were performed in the context of the 6th phase of the Coupled Model Intercomparison Project
(CMIP6; Eyring et al., 2016). These simulations cover the historical period – 1850 to 2014 – and a scenario period ranging
between the years 2015 and 2100.

The present study aims to investigate the high latitude response to increasing and decreasing atmospheric GHG concentra-
tions. As it is often easier to understand the underlying mechanisms when the effects are large, we investigate a high GHG
emission trajectory based on the Shared Socioeconomic Pathway 5 and the Representative Concentration Pathway 8.5 (SSP5-
8.5), even though this is not necessarily the most plausible scenario (van Vuuren et al., 2011; Riahi et al., 2017; Hausfather
and Peters, 2020). SSP5-8.5 targets a radiative forcing of 8.5 W m$^{-2}$ in the year 2100 and assumes the atmospheric $CO_2$
concentrations to increase to about 1000 ppmv by the end of this century, while the global mean temperature rises to about
4 K above pre-industrial levels and the precipitation in the high northern latitudes increases to about 675 mm year$^{-1}$ (Fig.
2). There are no scenario simulations available that could provide the forcing for a decrease in GHG concentrations. For the
present study, we thus assume that the decrease simply reverses the trajectory of the increase prior to the peak. We assess the
response to decreasing GHG concentrations after peaks in the years 2025, 2050, 2075 and 2100. It should be noted, however,
that these forcings are a simplification and do not necessarily provide the most realistic relation between GHG concentrations
and climate for the period of decreasing forcing as it ignores inertia in the climate system.

All simulations have the same general setup with a horizontal resolution of T63 ($1.9° \times 1.9°$), which corresponds to a grid-
spacing of about 200 km in tropical latitudes, a temporal resolution of 1800 seconds and a vertical resolution of 18 sub-surface
layers that reach to a depth of 100 m, 11 of which are used to represent the top 3 m of the soil column. Each simulation is
initialized in the year 1850 and the first 150 years of a simulation are used as a spin-up period. As stated above, the modi-
fications of the model introduce additional uncertainty and our strategy was not to choose the best estimate for many of the
parameters, but rather to vary them within the plausible range to capture the uncertainties that are involved in the respective
parametrizations. While a comprehensive analysis of these uncertainties is beyond the scope of this study, a concise overview
over the main factors is provided as an appendix.

A key factor determining the processes in the high latitudes is the treatment of the soil properties, especially those of the
organic fraction (Lawrence et al., 2008; Ekici et al., 2015; Jafarov and Schaefer, 2016; Zhu et al., 2019). For the study, we chose
2 configurations: One of these assumes a more loosely packed organic matter, e.g. a porosity of 85 % and a heat conductivity of
0.225 Wm$^{-1}$K$^{-1}$ for the dry organic matter below the surface, while the other assumes a denser organic matter, with a porosity
of 80 % and a heat conductivity of 0.275 Wm$^{-1}$K$^{-1}$. The soil properties are not only affected by the assumed properties of the
soil organic fraction, but also by the amount of organic matter stored in the soils. Here, it is extremely challenging to initialize
simulations with carbon pools that represent the observed organic matter concentrations adequately and we choose 4 sets of

initial soil carbon pools in which the amount of organic matter in high latitude soils ranges between 0.6 TtC and 0.9 TtC (see
below). Together with the assumed organic matter properties, this results in substantially different soil thermal and hydrologi-
cal properties and, consequently, substantial differences in the simulated sub-surface dynamics (Fig. 3 a-c). Another important
factor is the nitrogen limitation in the model. The changes in the hydrology module increase the leaching of mineral nitrogen
in the high latitudes, which reduces the nitrogen availability substantially. The corresponding nitrogen limitations are much
higher than in the standard model which has a drastic impact on the simulated vegetation dynamics and decomposition rates
(which are also limited by the nitrogen availability). Instead of re-tuning this highly uncertain parametrization we performed
an additional set of simulations in which the nitrogen limitations were neglected, capturing the range between potentially over-
and underestimated nitrogen limitations (Fig. 3 d-f).

Based on these configurations a core set ($n_{sim,core}$) of 16 simulations was performed, with:

$$n_{sim,core} = n_{c,props} \cdot n_{c,c-init} \cdot n_{c,nitro} = 16, \tag{28}$$

where $n_{c,props}$ is the number of configurations differing with respect to the soil properties (2), $n_{c,c-init}$ the number of configu-
rations with respect to the initial carbon pools (4) and $n_{c,nitro}$ the number of configurations with respect to nitrogen limitations
(2).

As Fig. 1 shows, the standard model has fundamental problems representing the dynamics in the high latitudes and we did
not include any simulations with the reference model version in our analysis. However, for 2 sets of simulations we reversed
certain key modifications to be as close to the standard model as possible. For one simulation we assumed vertically homo-
geneous soil properties and prevented the infiltration at sub-zero temperatures while allowing the supercooled water to move
vertically through the soil and to be used for microbial decomposition. In the second set we account for the impact of organic
matter on the soil properties only at the top of the soil column, while the lower layers have the properties of mineral soil. The
latter provides a model version that is very close to the setup used by Ekici et al. (2014, 2015).

Finally, for the region between $60°$ and $90°$ North our adapted model simulates present-day methane emissions of around 11
Mt(CH4), which is in good agreement with recent estimates of high latitude wetland emissions (Saunois et al., 2020). Nonethe-
less, we performed 2 additional sets of simulations in which we varied key parameters of the methane module to capture the
respective uncertainties. For one set of simulations we lowered the baseline fraction of $CH_4$ to 0.35 and increased the Q10
factor to 1.75, to represent the lower end of plausible methane emissions. For the other set – the high emission simulations –
we increased the baseline fraction of $CH_4$ to 0.45 and reduced the maximum oxidation velocities by 50 %.

In total, this gives 20 simulations ($n_{sim}$):

$$n_{sim} = n_{sim,core} + n_{c,pysics} + n_{c,CH_4} = 20, \tag{29}$$

where $n_{c,pysics}$ is the number of additional configurations with modified soil physics (2) and $n_{c,CH_4}$ the number of additional configurations in which the treatment of gases in the soil was modifed (2).

With respect to the results presented below, most of the analysis is performed based on aggregated values representative of the entire northern permafrost region – here defined as the areas that exhibit perennially frozen soils within the top 3 m of the soil column (Andresen et al., 2020). The extent of these areas is sensitive to the parameter values used in a specific setup and varies substantially between the simulations. For the analysis we do not define a shared permafrost mask, but aggregate the values based on the simulation-specific permafrost region. Furthermore, we base the analysis on the permafrost regions

at the beginning of the 21st century – roughly between 13 million $km^2$ and 16 million $km^2$ – and do not adjust their extent to account for the changes in the near-surface permafrost. Nonetheless, we simply refer to the focus region as the permafrost region in the manuscript even though large fractions of the respective areas may not feature near-surface permafrost at the higher temperatures of the assumed warming scenarios.

### 2.2.2   Initial carbon pools

Determining the initial soil carbon concentrations is very challenging, especially for the northern high latitudes where organic matter was stored in the frozen soils under the cold climate during and since the last glacial period (Zimov et al., 2006; Zimov, 2006b; Schuur et al., 2008). Simulations that target the build up of the soil carbon pools in permafrost-affected regions need to cover the carbon dynamics over a similar period (von Deimling et al., 2018). The respective simulations require many simplified assumptions and, because of the extensive timescale, even small uncertainties may propagate into substantial differences

between simulated and observed carbon pools. Another strategy is to initialize the simulations with observed soil carbon concentrations (Jafarov and Schaefer, 2016). These rely on the spatial extrapolations of thousands of soil profiles (Batjes, 2009, 2016; Hugelius et al., 2013) and can be considered much closer to reality than any modelling effort that we are aware of. However, this approach has the disadvantage that the carbon pools are not necessarily consistent with the simulated climate or with important boundary conditions used by the model (such as the soil depths), which can result in unrealistic carbon fluxes

especially at the beginning of a simulation. Furthermore, there is only little information on the quality of the soil organic matter, making it very difficult to separate the carbon into the lability classes used by the model. Here, we choose a combination of the two approaches to achieve some consistency with both observed soil carbon pools and the simulated climate.

To initialize the soil carbon concentrations, we mainly use the vertically resolved, harmonized soil property values from

the WISE30sec dataset (Batjes, 2016), which are based on soil profiles provided by the WISE project (Batjes, 2009, 2016). While this dataset only covers the top 2 meters of the soil column – other datasets provide information up to a depth of 3 m (Hugelius et al., 2014) – it has the important advantage that it is consistent with the FAO soil units which were used to derive the soil properties for the JSBACH model (Hagemann et al., 2009; Hagemann and Stacke, 2015). Consequently, we initialize the soil carbon concentrations above a depth of 2m with the WISE30sec data and for depth between 2 m and 3 m we

use data from the Northern Circumpolar Soil Carbon Database (NCSCDv2; Hugelius et al., 2014, see Fig. 4a,f). The datasets

provide no information on the quality of the organic matter and, for the most part, we distribute the soil carbon among the lability classes according to the pre-industrial equilibrium distribution that is simulated with the MPI-ESM. However, we do not assume the same distribution in all soil layers and make additional assumptions for different lability classes. The highly labile organic matter has a mass loss parameter that corresponds to a reference decomposition time ranging from a few days to a few years and the respective organic matter decomposes before it can be mixed throughout the soil column. Thus we assume that its vertical profile resembles that of the carbon inputs and distribute the highly labile carbon according to an idealized root profile. In contrast, the humus pool has a reference decomposition time of several hundred years, allowing it to be well mixed throughout the soil, and we assume a similar humus concentration in all layers.

$$C_{fast}(l) = TC_{obs} \cdot f_{fast,sim} \cdot y(l) \cdot dz(l)^{-1} \qquad\qquad \text{, with } TC_{obs} = \sum_{i=1}^{nlayers} C_{obs}(i) \cdot dz(i), \tag{30}$$

$$C_{slow}(l) = C_{obs}(l) \cdot f_{slow,sim}, \tag{31}$$

where $C_{fast}(l)$ is the concentration of highly labile carbon in layer $l$ and $C_{slow}(l)$ the humus concentration. $f_{fast,sim}$ and $f_{slow,sim}$ are the respective shares in the total soil carbon as simulated with the MPI-ESM. $C_{obs}(l)$ is the observed carbon concentration in layer $l$, $TC_{obs}$ the total amount of carbon in a given grid box, $y(l)$ the root fraction and $dz(l)$ the thickness of layer $l$.

In a first step we determine $C_{fast}$ with the above formula but apply the condition that wherever $C_{fast}(l) > C_{obs}(l)$, the excess in carbon is shifted to the nearest layer in which $C_{fast}(l) < C_{obs}(l)$. In a second step we calculate $C_{slow}$, iteratively applying the condition that wherever $C_{slow}(l) > (C_{obs}(l) - C_{fast}(l))$ the excess is added to the nearest layer in which $C_{slow}(l) < (C_{obs}(l) - C_{fast}(l))$. After $C_{fast}(l)$ and $C_{slow}(l)$ have been determined, $C_{med}(l)$, the concentration of organic matter with a decomposition timescale of tens of years, is calculated as the difference between $C_{obs}(l)$, and the sum of $C_{fast}(l)$ and $C_{slow}(l)$:

$$C_{med}(l) = C_{obs}(l) - (C_{fast}(l) + C_{slow}(l)). \tag{32}$$

As stated above, there are regions in the high latitudes where the observed carbon pools are not only inconsistent with the simulated climate but even with the soil depths used by the model. The main reason for this is the high spatial variability in soil depths in the real world which can not be represented at the coarse resolution of the model, resulting in large amounts of the observed organic matter being stored in parts of the ground that the model considers to be below the bedrock boundary. When limiting the organic matter to the top- and subsoil – the soil above the bedrock – of the standard setup, the model is initialized with as little as 636 GtC of organic matter in the northern high latitudes instead of the observed 1015 GtC (Fig. 4a,b,f,g). Consequently, limiting the initial carbon pools to the observed organic matter concentrations includes the risk of substantially underestimating the effect of increasing temperatures on the soil carbon release.

There are two general approaches to mitigate this problem, both of which introduce different risks for the setup of the simulation. One approach is to upscale the carbon pools that are located above the bedrock boundary to obtain an overall carbon

content that is closer to observations (Fig. 4c,h). However, this approach introduces the risk of partly overestimating the organic matter concentrations and misrepresenting the soil properties in the respective regions. The second approach extends the soil depths in the model (Fig. 5; Carvalhais et al., 2014; von Deimling et al., 2018), which allows to store more organic matter – about 797 GtC – in the appropriate regions (Fig. 4d,i). These soil depths, however, do not represent the bedrock boundary appropriately at coarse resolutions, which may strongly affect the behaviour of the model in these regions. Finally, the two approaches can be combined by upscaling the carbon pools while simultaneously increasing the soil depths of the model (Fig. 4e,j). Because none of the approaches can solve the fundamental problem of subgrid-scale heterogeneity in a coarse resolution model, we conducted 4 sets of simulations with all the above initialization approaches and a short overview over the effect on the simulated carbon dynamics is provided in the appendix.

To minimize the inconsistency between observed carbon concentrations and simulated climate conditions we initialize the experiments with the observation-based, present-day carbon pools but start the simulations in the year 1850 at the end of the pre-industrial period. In the high northern latitudes this allows the carbon concentrations within the (simulated) active layer to adapt to the simulated climate conditions during the historical period, while the perennially frozen regions of the soil conserve the observed carbon concentrations. As the soil thermal and hydrological dynamics vary depending on the treatment of the soil properties, this initialisation approach results in substantially different soil carbon pools at the end of the spin-up period.

## 3 Results

At the beginning of the 21st century, permafrost regions (Fig. 3a) contain between 373 and 764 Gt of organic carbon (Tab. 1). 171 to 298 GtC of these are located within the active layer, where the organic matter is exposed to microbial decomposition, and the resulting soil $CO_2$ emissions range between 2.4 and 4.0 GtC year$^{-1}$. In most simulations these emissions are fully balanced by the carbon uptake of the soil, resulting in net fluxes of between -0.2 and 0.5 GtC year$^{-1}$. This also is the case for the terrestrial ecosystem as a whole (Fig. 6 a) – when also accounting for changes in vegetation biomass – and the simulated ecosystem carbon flux into the atmosphere at the beginning of the 21st century ranges between -0.8 and 0.1 GtC year$^{-1}$. However, the ecosystem flux increases substantially with rising temperatures and for the the Paris Agreement longterm goal – global mean surface temperatures limited to about 1.5 K above pre-industrial levels – the simulated net fluxes increase from -0.3 GtC year$^{-1}$ to around -0.1 GtC year$^{-1}$. At temperatures of roughly 1.75 K ($\pm$ 0.5 K) above pre-industrial levels the permafrost ecosystem turns from carbon sink to source and for a temperature rise of 3 K the net emissions increase to about 1 GtC year$^{-1}$. The ecosystem emissions exhibit a non-linear, hysteretic dependence on the simulated surface temperatures and the fluxes into the atmosphere are substantially lower after the GHG peaks in 2050, 2075 and 2100 than before. Here, the ecosystem flux is largely determined by $CO_2$ exchange between the land – soil and vegetation – and the atmosphere, while methane emissions contribute very little to the overall carbon flux (Fig. 6 b,c).

## 3.1 Soil CO$_2$ flux and carbon uptake

The CO$_2$ emissions from permafrost-affected soils are very sensitive to changes in the atmospheric GHG concentration. A substantial increase in soil CO$_2$ fluxes becomes very likely should 21st-century GHG emissions follow the SSP5-8.5 scenario. The fluxes depend on a number of factors that are affected by the atmospheric GHG concentrations, most importantly the changing climate conditions. As a result, the soil emissions exhibit a non-linear dependence on the atmospheric CO$_2$ concentrations (not shown) but an almost linear dependence on the simulated surface temperatures (Fig. 6 b) – where, very roughly, each degree of global warming increases the soil CO$_2$ fluxes by 50%, relative to the emissions at the beginning of the century. For the temperature target of the Paris Agreement, the soil emissions increase by about 25% to 40%. If the GHG concentrations follow SSP5-8.5 until the year 2100, the soil CO$_2$ fluxes potentially increase by more than 150%, resulting in fluxes of roughly 6 to 11.5 GtC year$^{-1}$. The (almost) linear temperature dependence of the soil CO$_2$ fluxes is also valid for decreasing temperatures and the soil CO$_2$ fluxes decrease on a trajectory very similar to the increase prior to the GHG peak when the forcing is reversed. However, this does not necessarily mean that the main processes governing the changes in soil CO$_2$ emissions are fully reversible on a decadal to centennial timescale (see below).

One reason for the strong increase in soil CO$_2$ fluxes with rising temperatures is the degradation of near-surface permafrost and the corresponding increase in active layer depth. For a forcing peak in 2100, over 80 % of the near-surface permafrost disappear (Fig. 7 a), exposing large amounts of organic matter to conditions which permit microbial decomposition. However, the amount of soil organic matter in permafrost-affected regions actually increases as long as the global mean surface temperature remains below 1.5 K compared to pre-industrial levels. Only for higher temperatures does the rise in soil respiration, resulting from the increased exposure of formerly frozen carbon, reduce the soil organic matter. For a forcing peak in 2100 the total amount of carbon stored in permafrost-affected soils could be reduced by about 12.5 % (Fig. 7 b). This corresponds to a loss of roughly $60 \pm 20$ GtC, which is at the lower end of previous estimates (Schuur et al., 2013; Schaefer et al., 2014). Because the soils start to accumulate organic matter when the forcing is reversed, the soil carbon pools increase after the GHG peak, and for most of the scenarios the soil carbon concentration increases again, at least to the levels of the beginning of the simulation.

The increased exposure of organic matter stored in permafrost soils is insufficient to explain the rise in soil CO$_2$ fluxes, especially when considering that the soil carbon concentration initially increases, as long as temperatures stay below 1.5 K compared to pre-industrial levels. Furthermore, the largest fraction of the recently exposed organic matter takes a long time to decompose (Fig. 7 c), while the relative increase in readily decomposable material within the active layer is much smaller (Fig. 7 d). Here, the amount of labile active layer carbon starts to decrease even before the forcing peak in 2100 is reached, while the soil CO$_2$ fluxes continue to increase. Furthermore, the labile carbon in the active layer shows a strongly hysteretic behaviour and after the forcing peak there is substantially less labile organic matter in the active layer than prior to the peak, while there is slightly more stable organic matter. This indicates that the permafrost-affected soils undergo major compositional changes

which should lead to lower soil $CO_2$ fluxes after the temperature peak than before.

Especially at lower temperatures, the main driver of the soil $CO_2$ fluxes is the carbon input into the soil, consisting of litter, root exudates but also damaged and burnt vegetation, which is largely dependent on the net primary productivity (NPP). The NPP, in turn, depends directly on the atmospheric GHG concentrations, as this determines the $CO_2$ uptake by leaves. Furthermore it depends on $CO_2$ indirectly, through the resulting climate conditions, namely surface temperatures and water availability, as well as the vegetation distribution, which is characterised by the type of vegetation and the vegetation cover. With surface temperatures in the Arctic increasing about twice as fast as the global mean, the high latitudes become much more habitable for plants (Fig. 8 a). Higher temperatures extend the growing season, as well as increasing the water availability for plants, because higher soil temperatures cause the soil ice to melt earlier and refreeze later during the year. Together with the increase in precipitation (Fig. 2 d) this raises the plant-available water by up to 50 % (Fig. 8 b). Furthermore, the changes in climate conditions also increase the vegetation cover and facilitate a (relative) shift from grasses and shrubs towards more productive trees (Fig. 8 c). In combination with the direct effect of $CO_2$ fertilization, the changes in climate and vegetation increase the NPP in permafrost-affected regions substantially, more than doubling the productivity at the time of the GHG peak in 2100 (Fig. 8 d).

This increase in NPP corresponds to an increase in carbon input into the soil of up to 3.5 GtC year$^{-1}$ and as long as temperatures stay below 1.5 K above pre-industrial levels, the rise in soil $CO_2$ emissions is (more than) balanced by the increase in the soil carbon inputs. Even for the temperature peak in 2100 about half of the increase in soil $CO_2$ fluxes is balanced by the increase in primary productivity. After the GHG peak the NPP is consistently larger than before the peak, mostly because the tree cover remains very high, resulting in substantially larger carbon inputs than prior to the peak. This balances the reduced amount of labile carbon in the active layer, explaining why soil $CO_2$ emissions are very similar before and after the temperature peak, despite the reduced availability of labile carbon in the active layer – by up to 25% – during the temperature decrease. Thus, the predominant absence of hysteresis in the simulated soil $CO_2$ emissions does not mean that the governing processes are fully reversible on short timescales, but it rather is the result of two strongly hysteretic factors offsetting each other. Before the GHG peak the large $CO_2$ fluxes are supported by the deepening of the active layer, while it is largely the increase in NPP that drives the post-peak $CO_2$ fluxes. The larger carbon uptake by plants following the GHG peak also explains the hysteresis of the ecosystem net carbon flux. With similar soil $CO_2$ fluxes and a substantially larger NPP, the flux into the atmosphere is consistently smaller after the GHG peak than prior to it.

Here, it should be noted that the hysteretic behaviour arises partly because the characteristic timescales of the high latitude carbon cycle, most importantly of vegetation shifts and the decomposition of soil organic matter, are larger than the timescales of the climate change scenarios investigated. In addition, high latitude soils have a large thermal inertia, especially due to the large amounts of energy required or released by the phase change of water within the ground. Thus, the simulated behaviour does not necessarily indicate the multistability of the system, but may merely exhibit a transient hysteresis as described by

Eliseev et al. (2014). However, the question whether the hysteresis is purely transient or indicative of multistability is beyond the scope of this study and the subject of an ongoing investigation.

## 3.2 $CH_4$

The methane emissions from permafrost-affected soils behave very differently from the soil $CO_2$ fluxes. Most importantly, the soil $CH_4$ emissions are 3 orders of magnitude smaller than the respective $CO_2$ fluxes, indicating that methane will play only a minor role in the northern high latitude contribution to global warming, even when considering the respective difference in global warming potential. At the beginning of the 21st century the simulated net $CH_4$ emissions from high latitude soils amount to roughly 7 MtC year$^{-1}$ – or about 9 Tg($CH_4$) year$^{-1}$. With a global warming potential of 28 times that of $CO_2$ (Stocker et al., 2013), this corresponds to a $CO_2$ flux of 0.2 GtC year$^{-1}$. The spread in the simulated methane fluxes is substantial, however even the largest present-day net $CH_4$ emission of any of the simulations is below 25 MtC year$^{-1}$, which has the warming potential of a $CO_2$ flux of 0.7 GtC year$^{-1}$.

One reason for the low soil $CH_4$ fluxes produced in the anaerobic decomposition of organic matter is the temperature dependence that determines the ratio of $CH_4$ and $CO_2$. For the low temperatures that are characteristic for the high northern latitudes, only a small fraction – on average around 20% – of the anaerobically decomposed organic matter is converted into methane. Furthermore, the area in which anaerobic conditions occur is comparatively small. The vast majority of all inundated areas are flooded only seasonally and anaerobic conditions in the soil only exist temporarily, predominantly during late spring and early summer (Fig. 3 g-i). Thus, while there are regions in western Siberia where up to 40% of the surface are inundated during the snow-melt season, the average inundated fraction in permafrost-affected regions ranges roughly between 4% and 6%, which increases to about 10% to 12% when only considering the period of April - June. On one hand this means that the amount of organic matter that is decomposed under anaerobic conditions is roughly an order of magnitude smaller than the amount decomposed under aerobic conditions – around 0.1 GtC year$^{-1}$ compared to 3 GtC year$^{-1}$. On the other hand, it means that the largest fraction of the high latitude soils produces no methane but actually takes up atmospheric $CH_4$, oxidising it to $CO_2$.

The way the $CH_4$ emissions react to changes in atmospheric GHG concentrations also differs substantially from the $CO_2$ fluxes (Fig. 6 c). The relative increase with rising GHG levels is substantially smaller and the emissions even start to decrease when global mean surface temperatures rise beyond 3 K above pre-industrial levels. This is very different from the results of previous modelling studies, who found a strong positive connection between the 21st century temperature rise and methane emissions (Khvorostyanov et al., 2008a; Burke et al., 2012; von Deimling et al., 2012; Schuur et al., 2013; Lawrence et al., 2015). In large parts, the behaviour of the simulated methane emissions is a result of the dependence of the net $CH_4$ fluxes on the atmospheric methane concentrations, as the former are determined by the $CH_4$ gradient between the soil and the atmosphere. The SSP5-8.5 scenario predicts the atmospheric methane concentrations to increase by more than 40% by the end of the 21st century (Fig. 2 b). Consequently, the methane concentrations in the soil have to increase similarly, merely to maintain constant $CH_4$ fluxes. The same is true for the $CO_2$ fluxes, however there is an important difference: when the soil-atmosphere

$CO_2$ gradient decreases due to increasing atmospheric GHG concentrations, $CO_2$ rapidly accumulates in the soil until an equilibrium with the atmospheric concentrations is reached and further $CO_2$ generated by the decomposition of soil organic matter will be released to the atmosphere. In contrast, it is much more difficult for the $CH_4$ concentrations to build up within the soil because a large fraction of the methane is constantly converted into $CO_2$ in the oxygen-rich soil layers near the surface. Additionally, larger atmospheric methane concentrations increase the $CH_4$ flux into non-saturated soils that do not produce methane. For larger atmospheric GHG concentrations, the high northern latitudes could even act as a net methane sink despite rising temperatures increasing the $CH_4$ production within the soil.

However, the atmospheric $CH_4$ concentration can not explain why the methane fluxes are substantially smaller after the forcing peak and why the high latitude soils may remain a methane sink when the forcing is fully reversed and atmospheric GHG concentrations have returned to present-day levels. The highly hysteretic behaviour of the methane emissions is the result of changes in the methane production in the soil and the way methane is transported towards the surface. Soil respiration depends on the availability of organic matter and the decomposition rates. The latter are determined by the conditions under which the organic matter decomposes. These conditions are not only affected by changes in near-surface climate but also vary depending on the (vertical) position of the organic matter within the soil column, making the soil $CH_4$ emissions strongly dependent upon changes in the vertical soil carbon profile.

The most important effect of rising GHG concentrations is an increase in soil and surface temperatures. While rising temperatures have a predominantly positive effect on soil respiration, due to the temperature dependence of the decomposition rates, they can be detrimental to the occurrence of inundated areas and saturated soils, hence the areas where organic matter is decomposed under anaerobic conditions. On one hand rising temperatures increase evapotranspiration in summer which decreases the inundated areas after the spring snow melt. On the other hand, ice within the deeper layers of the soil acts as a barrier and the soils drain much more readily when this barrier is melted. Climate warming also increase precipitation rates by up to 25% (Fig. 2 d), partly balancing the negative effect that higher temperatures have on the extent of inundated areas. The combined effect of increasing temperatures and precipitation rates is a slight drying of the soils, which has also been found by other models (Andresen et al., 2020). However, this drying of the soil has only a small impact on the overall extent of inundated areas (Fig. 9 a). This is because the spatial distribution of inundated areas adapts to the changes in climate conditions, with their extent decreasing in lower and increasing in higher latitudes (Fig. 10). Furthermore, the liquid soil water content increases substantially in regions that feature large wetland areas (Fig. 9 b), while the overall water content (including soil ice) declines slightly (not shown).

With a similar extent in the inundated area and increasing temperatures, the soil methane production mainly benefits from the changes in climate. However, the oxic soil respiration in the adjacent non-inundated areas increases even more because, in addition to the effects of rising temperatures, the increase in liquid water content reduces the moisture limitations on the oxic decomposition rates (Fig. 9 c,d). Furthermore, the vertical distribution of organic matter in the soil changes in a way that also

increases the $CO_2$ production relative to that of $CH_4$. The largest fraction of the carbon inputs occurs above or close to the surface. Thus the increase in NPP, resulting from rising GHG concentrations, primarily increases the carbon concentrations at the top of the soil column (Fig. 11 a, b). At the surface the oxic decomposition rates are much larger than those under anoxic conditions, while this difference is less pronounced deeper within the soil. Consequently, the (relative) increase in organic matter at the surface further increases the difference between the oxic and anoxic respiration rates. When the forcing is reversed, the factors that determine the difference between the decomposition rates do not return to their state prior to the forcing peak – the distribution of inundated areas remains very different (Fig. 10), the liquid water content in the soil remains much higher (Fig. 9 b) and the soil carbon profile still exhibits higher concentrations of organic matter closer to the surface (Fig. 11 c). Consequently, the difference between oxic and anoxic decomposition rates are also larger after the forcing peak than prior to it.

The difference in the decomposition rates is highly relevant for the soil methane production. The largest fraction of the anaerobic decomposition takes place in seasonally saturated soils, which means that the organic matter in the respective areas decomposes under aerobic conditions for a given period. Consequently, the soil methane production depends on both, the anoxic and the oxic decomposition rates, as the latter determine how much organic matter is decomposed during the drier months, hence how much organic matter is available at the onset of inundation.

The inundated area is largest during spring and early summer, April to July, with a peak in May and June followed by a sharp decline due to a strong increase in evapotranspiration (Fig. 12 a). Productivity, on the other hand, peaks in July and the decomposition rates and the litter flux peak even later. Thus, a large fraction of the carbon input into the soils occurs when conditions favour oxic over anoxic decomposition. Increasing temperatures strengthen this effect because the extent of inundated soils increases during spring – due to larger snow melt fluxes – while it decreases during summer, owing to higher evapotranspiration and drainage rates (Fig. 12 b). Most importantly, the increase in oxic decomposition rates is far larger than the increase in anoxic decomposition rates. Thus when GHG concentrations increase, the largest fraction of the additional organic matter, resulting from the increased productivity and the deepening of the active layer, is being respired under oxic conditions in summer and fall when the extent of inundated areas is relatively small. Additionally, higher soil temperatures lead to a larger fraction of the fresh litter being decomposed aerobically during winter, resulting in a lower soil carbon availability during and after the following snow melt season. Thus, for large temperature increases, the relative increase in oxic soil respiration in regions that feature large wetland areas is much more pronounced than the relative increase in methane production (Fig. 9 c,d).

When the climate forcing is reversed, there is less organic matter within the active layer, especially during summer and fall, but the decomposition rates and the litter flux remain higher than prior to the peak (Fig. 12 c). In the case of aerobic soil respiration the increase in decomposition rates is so large that the soil respiration during winter and spring is actually larger than before the forcing peak, despite the reduced availability of organic matter. This partly balances the reduction in soil respiration during summer and fall and the annually averaged aerobic soil respiration following the forcing peak is only slightly smaller

than prior to the peak. In contrast, the anoxic decomposition rates increase just enough to balance the reduced soil carbon availability during the winter and spring months, while there is substantially less anoxic soil respiration during summer and fall. In the case of anaerobic respiration, the effect of the lower soil carbon availability is even more severe because more organic matter, including the litter input, has been respired aerobically during the previous dry period. Thus, because the anaerobic decomposition rates are much lower than the aerobic rates, while both anoxic and oxic respiration (partly) depend on the same carbon pools, the relative changes in oxic and anoxic respiration are very different. In regions that feature large inundated areas, the oxic respiration is only a few percent lower after than before the forcing peak, while the soil methane production is reduced by up to 30% (Fig. 9 c,d).

Still, even the reduced methane production can not fully explain the difference in the net $CH_4$ emissions before and after the forcing peak. Another important factor is the methane transport in the soil and how this is affected by the changes in the vertical soil carbon profile. There are two major pathways by which methane is transported towards the surface, one being the diffusive transport through the soil, the other being plant-mediated transport. The model also simulates ebullition, but the respective fluxes can be neglected. Even in saturated soils the layers near the surface have a high oxygen content and the better part of the methane that diffuses upwards is oxidised within these layers. Consequently, the largest methane fluxes at the surface do not result from vertical diffusion, but from the methane release by (vascular) plants, whose roots absorb methane within the soil and transport it to the atmosphere via aerenchyma. Here, the changes in the vertical soil carbon profile alter the fraction of methane that is transported towards the surface by a given transport mechanism. When the share of organic matter increases close to the surface and decreases in the deeper layers, a smaller fraction of the methane produced in the soil can be absorbed by roots, substantially reducing the respective emissions at the surface. In a addition there is a 5% decrease in the cover fraction of grasses, which are the most effective gas transporters (not shown). As a result, the methane emissions by plants decrease by up to 60% when the climate forcing is fully reversed, which is a reduction roughly twice as large as the relative decrease in the soil methane production (Fig. 13 a). In contrast, the oxidation rates differ comparatively little before and after the forcing peak (Fig. 13 b) because the lower methane concentrations in the soil lead to a larger uptake of atmospheric $CH_4$. Thus, with a reduced methane production and a larger $CH_4$ uptake the net emissions from the soil decrease substantially, potentially turning the permafrost-affected regions into a net $CH_4$ sink.

## 4   Conclusions

One of the most important factors in the permafrost-carbon-climate feedback is the fraction of soil carbon that is released in the form of $CH_4$, which is tightly connected to the question of whether the high latitudes will become wetter or drier in the future (Schuur et al., 2015). Here, many land surface models indicate a drying of the high latitudes (Andresen et al., 2020) which will most likely constrain the decomposition under anaerobic conditions (Oberbauer et al., 2007; Olefeldt et al., 2012; Elberling et al., 2013; Schaedel et al., 2016; Lawrence et al., 2015). In parts, this is confirmed by results of our study and while we did not find a substantial decrease in the extent of areas with saturated soils, we also did not find a significant expansion of

the inundated areas in high latitudes, despite the pronounced increase in precipitation resulting from the SSP5-8.5 scenario. In addition, there is a distinct spatial shift in the wetland area, with the extent decreasing in the more southerly and increasing in the more northerly permafrost regions. This shift limits the methane production with increasing temperatures – as organic rich soils are predominantly located in the more southerly regions – and is not fully reversible on decadal timescales. Furthermore, we could show that the high latitude methane fluxes are strongly limited by the increase in oxic decomposition because most

of the soils are saturated only seasonally and the availability of organic matter depends on the respiration during the drier months of the year. But, most importantly, we found the methane oxidation in the soil to be the dominant constraint on the soil $CH_4$ emissions. Even for present conditions, less than half of the methane produced in permafrost-affected soils was actually emitted at the surface. With the atmospheric $CH_4$ concentration increasing and the vertical methane transport by vascular plants decreasing, this fraction could be reduced to less than a third in the future. Because of these limitations, there was not

a single year – from 10000 years of simulation (500 years for 20 ensemble members) – in which the methane emissions from permafrost-affected soils exceeded 50 MtC year$^{-1}$. Considering the global warming potential of methane, this corresponds to a $CO_2$ flux of about 1.4 GtC year$^{-1}$, which is an order of magnitude smaller than the largest simulated $CO_2$ flux – about 13 GtC year$^{-1}$.

Thus, our results indicate that the soil methane fluxes in permafrost-affected regions do not constitute an important contributor to the climate-carbon feedback. Here, it should be noted that the net emissions could be even lower as a recent study has indicated that the methane uptake in dry soils could be severely underestimated due to the omission of recently identified high-affinity methanotrophs (Oh et al., 2020), especially under future climatic conditions. In contrast, the soil $CO_2$ emissions are so large that the (terrestrial) Arctic ecosystem turns into a source for atmospheric carbon when temperatures increase beyond

1.75 K ($\pm$ 0.5 K) above pre-industrial levels. By the end of the 21st century the net ecosystem emissions could increase to up to 2 GtC year$^{-1}$, which not only places them on par with present-day land use change emissions but would also substantially reduce the overall terrestrial carbon uptake (Le Quéré et al., 2018). This is very different from scenario simulations with the standard version of the MPI-ESM1.2 in which the region continues to take up atmospheric $CO_2$ throughout the entire 21st century, with the net uptake increasing from about 0.005 GtC year$^{-1}$ to around 0.015 GtC year$^{-1}$ (not shown). These differences

confirm that the non-consideration of permafrost-related processes and the organic matter stored in the frozen soils leads to a fundamental misrepresentation of carbon dynamics in the Arctic.

Despite their importance, the processes governing the carbon dynamics in permafrost-affected regions are not fully taken into account in the present generation of Earth system models. Substantial advances have been made within the last decade –

many land surface models now include some physical and biogeochemical permafrost processes (McGuire et al., 2016; Chadburn et al., 2017). However, hardly any of the models that participated in CMIP6 included an adequate representation of the soil physics in high latitudes, while simulating (interactive) vegetation dynamics as well as the carbon and nitrogen cycle. Consequently, model-based studies, at present, can merely provide qualitative answers to the question how permafrost thaw may contribute to global warming (Schuur et al., 2015). Even when the most relevant processes are included, there are large

uncertainties regarding the respective parametrizations, as well as the initial- and boundary conditions used by the model, many of which are only poorly constrained by observations. In the present study we used an ensemble of simulations in which we varied key parameters within the uncertainty range and while the simulations largely agree on the relative response of the system to increasing and decreasing GHG concentrations, the spread in the absolute values was substantial between the ensemble members, e.g. the simulated carbon pools at the beginning of the 21st century ranged between 373 and 764 GtC, while the soil

methane emissions ranged between 2.8 and 19.3 MtC year$^{-1}$. To be able quantify the impact of permafrost degradation on the climate system, these uncertainties need to be reduced substantially.

*Code and data availability.* The primary data is available via the German Climate Computing Center long-term archive for documentation data (https://cera-www.dkrz.de/........, to be specified before publication). The model, scripts used in the analysis and other supplementary

information that may be useful in reproducing the authors' work are archived by the Max Planck Institute for Meteorology and can be obtained by contacting publications@mpimet.mpg.de.

*Author contributions.* P.d.V designed experiment, performed model adaptation, conducted simulations and analysis, T.S. and T.K performed model adaptation and conducted parts of the analysis and V.B. was involved in experiment design and conducted parts of the analysis. All authors reviewed the manuscript.

*Competing interests.* The authors declare that they have no competing financial interest.

*Acknowledgements.* This work was funded by the German Ministry of Education and Research as part of the KoPf-Project (BMBF Grant No. 03F0764C).

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

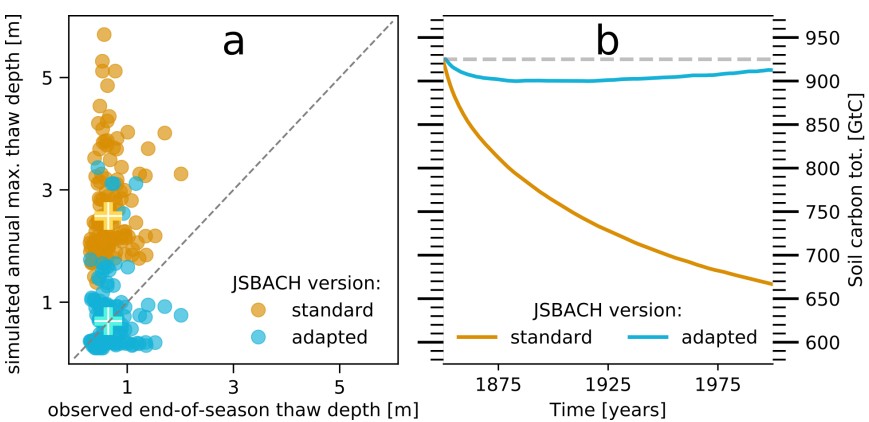

**Figure 1. JSBACH model – standard and adapted version:**

**a)** Correlation between simulated annual maximum and observed end-of-the-season thaw depths for the sites of the Circumpolar Active Layer Monitoring (CALM; Brown et al., 2000) program that are located in the simulated permafrost domain (Fig 3). Brown dots indicate the maximum thaw depths simulated with the JSBACH standard model version, while blue dots refer to the adapted version used in the present study. The CALM dataset encompasses the period from 1990 to present. However, this is not the case for all the included sites and the individual dots show the mean over the period covered by data at a specific site. Crosses show the average over all sites for the respective model versions. Finally, it should be noted that the simulations were performed with the soil properties at the standard resolution ($1.9° \times 1.9°$) and with atmospheric conditions from a historical simulation with the MPI-ESM, which may be different from the actual soil properties and meteorological conditions at the specific sites. **b)** Simulated soil carbon in the permafrost domain during the historical period. The brown line refers to the standard JSBACH model, while the blue line shows the simulation with the adapted version. The grey dashed lines shows the observation-based soil carbon stocks (Fig. 4e), with which the simulations were initialized.

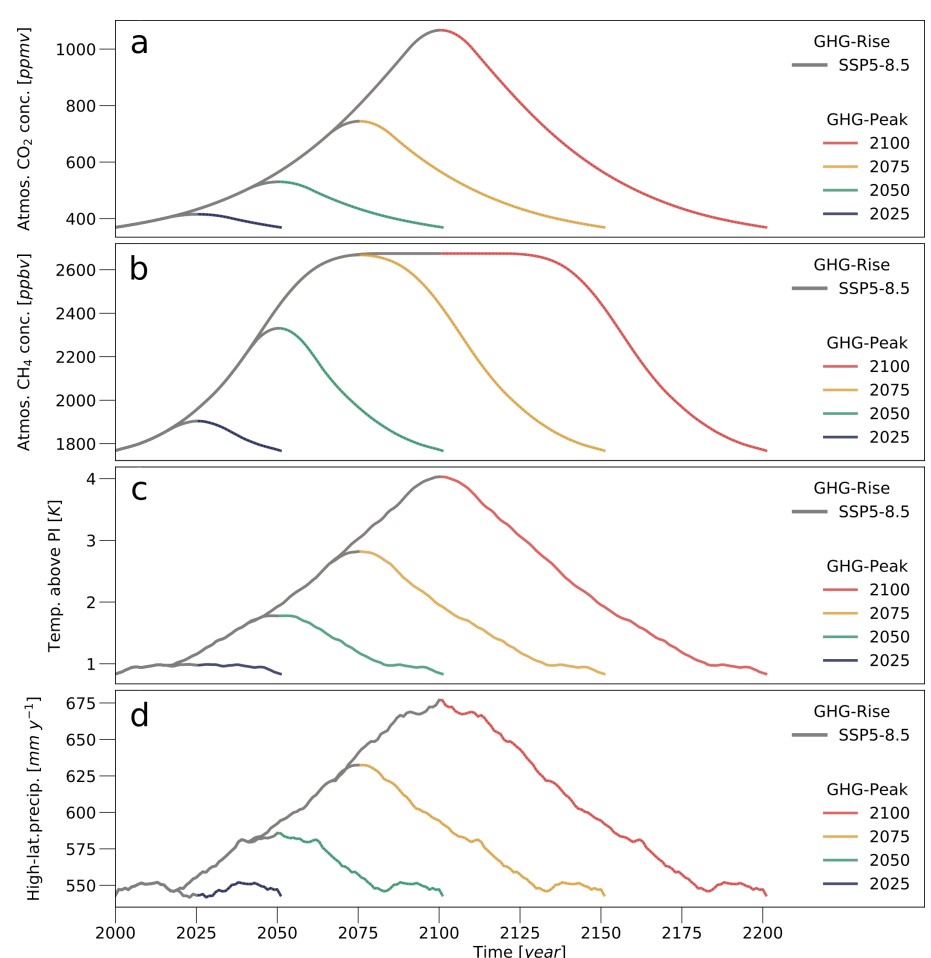

**Figure 2. Experimental setup:** Forcing used after the 1850 - 2000 spin up phase: **a)** Atmospheric $CO_2$ concentrations. **b)** Atmospheric $CH_4$ concentrations. **c)** Global mean surface temperature relative to the pre-industrial temperature. Note that the model is not forced by surface temperatures directly, but by atmospheric temperatures at a height of roughly 30 m and the surface incoming long- and short-wave radiative fluxes. **d)** Precipitation rates, averaged over the latitudinal band between 60° and 90° North. Grey lines show the forcing according to the SSP5-8.5 scenario. The coloured lines show the forcing-pathways that are used to reverse the forcing to the state at the beginning of the 21st century – after an assumed peak in the year 2025 (blue), 2050 (green), 2075 (yellow) and 2100 (red). In case of temperature (and the surface radiative fluxes) and precipitation rates, the forcing was derived from CMIP6 scenario simulations with the fully coupled MPI-ESM. All panels show the 20-year moving average of the respective variable.

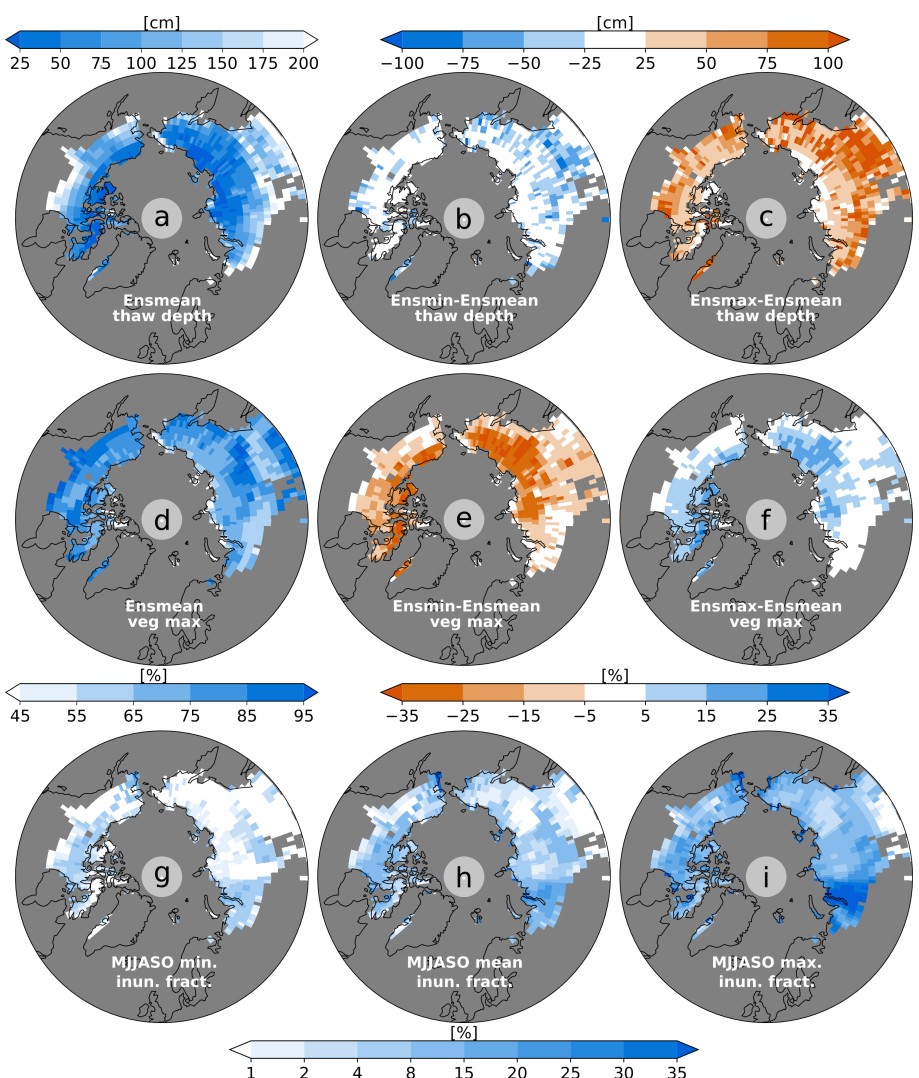

**Figure 3. Simulated permafrost, vegetated fraction and inundated areas in the year 2000: a)** Ensemble mean of the annual maximum thaw depth, corresponding to the year 2000 (1990 - 2010 mean). Grey areas indicate grid boxes in which the annual maximum temperatures throughout the top 3 m of the soil exceeded the melting point for more than 10 years in the period 1990 - 2010. These are considered to be unaffected by near-surface permafrost and are not taken into consideration in the study. **b)** Same as a but for the difference between ensemble minimum and mean. **c)** Same as a but for the difference between ensemble maximum and mean. **d)** Ensemble mean vegetated fraction. Note that this is the maximum grid box fraction that can be covered by vegetation, while the actual vegetated cover depends on the current state of the vegetation and can vary throughout the year. **e)** Same as d but for the difference between ensemble minimum and mean **f)** Same as d but for the difference between ensemble maximum and mean. **g)** Minimum inundated fraction during the summer months (May-October) for the year 2000. Shown is the ensemble mean. **h)** Same as g but for the May-October mean. **i)** Same as g but for the May-October maximum inundated fraction.

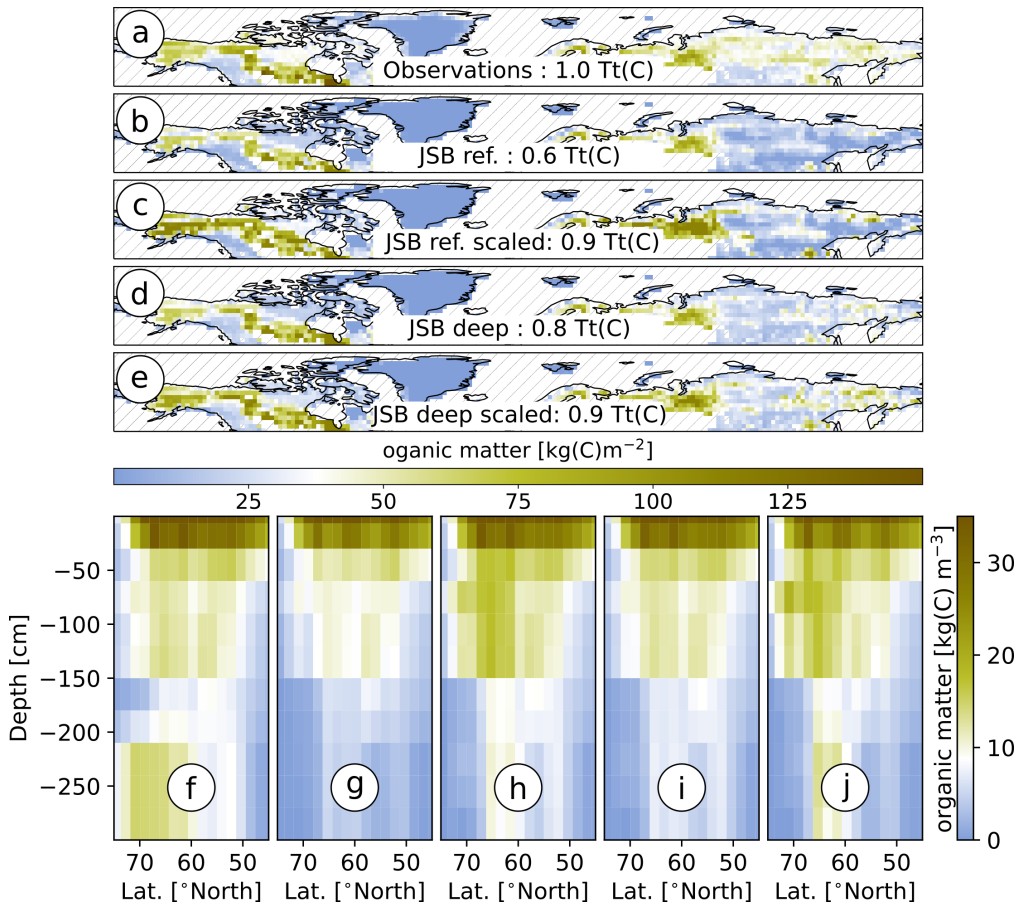

**Figure 4. Initial soil carbon**

**a)** Spatial distribution of soil organic matter in permafrost regions based on WISE30sec data, above a depth of 2 m, and on NCSCDv2 data for depths between 2 m and 3 m. In regions in which NCSCDv2 is available – roughly those that, in reality, are affected by continuous, discontinuous and sporadic permafrost – the soils contain about 1015 Gt organic carbon. **b)** Same as *a* but only the organic matter that is located above the bedrock border assumed in the standard model setup – roughly 636 GtC. **c)** Same as *b* but for up-scaling those carbon pools that are located above the bedrock border to 858 GtC. **d)** Same as *b* but for a setup that assumes deeper soils – 797 GtC. **e)** same as *c* but for a setup that assumes deeper soils – 931 GtC. **f)** same as a but showing the vertical distribution (zonal average). **g)** same as b but showing the vertical distribution. **h)** same as c but showing the vertical distribution. **i)** same as d but showing the vertical distribution. **j)** same as e but showing the vertical distribution.

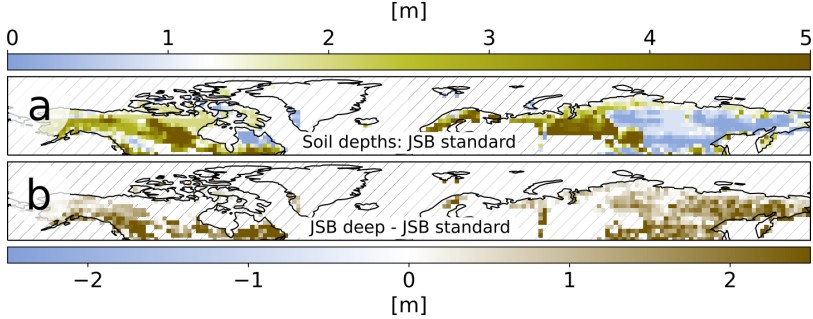

**Figure 5. Soil depths**

a) Soil depths used in the JSBACH standard setup. b) Difference in soil depths between the setup with deeper soils (Carvalhais et al., 2014; von Deimling et al., 2018) and the standard setup.

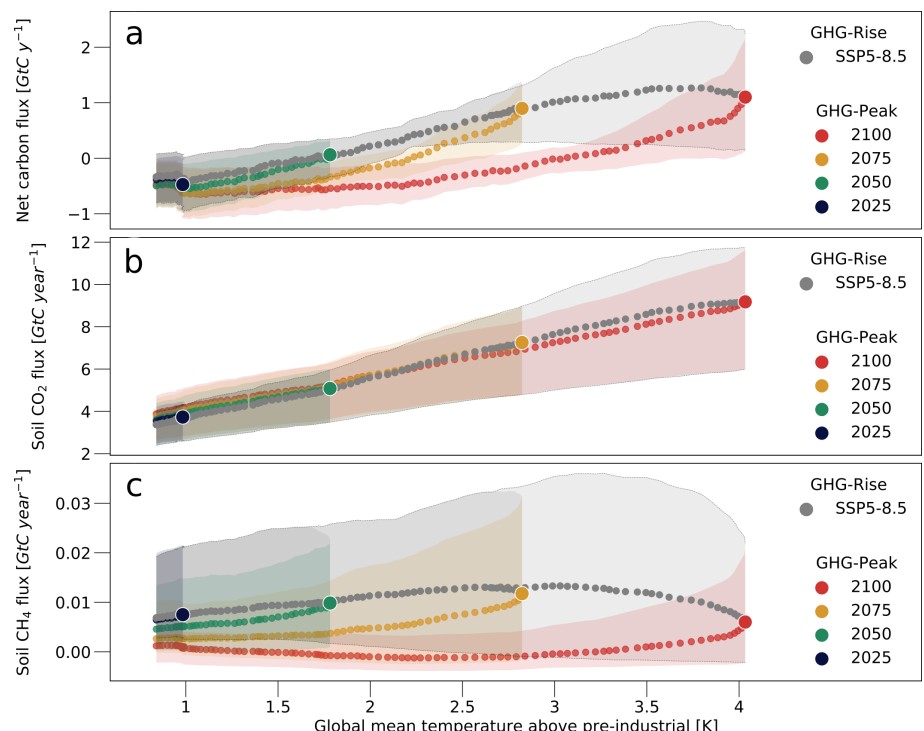

**Figure 6. Net ecosystem carbon flux, soil CO$_2$ flux and soil methane emissions in permafrost-affected areas: a)** Simulated net CO$_2$ flux into the atmosphere, taking into account heterotrophic and autotrophic respiration, disturbances, land-use emissions and the CO$_2$ uptake by plants. **b)** Same as a but showing the soil CO$_2$ emissions. **c)** Same as a but showing the soil (net) methane flux. Grey dots show the ensemble mean increase in emissions as a function of the temperature increase according to the SSP5-8.5 scenario. Coloured dots indicate the ensemble mean decline in fluxes for the reversion of the forcing after a forcing-peak in 2025 (blue), 2050 (green), 2075 (yellow) and 2100 (red). Each dot represents a 20 year (moving) average. Shaded areas indicate the spread between the ensemble minimum and maximum. The figure is representative of those areas that were affected by near-surface permafrost in the year 2000.

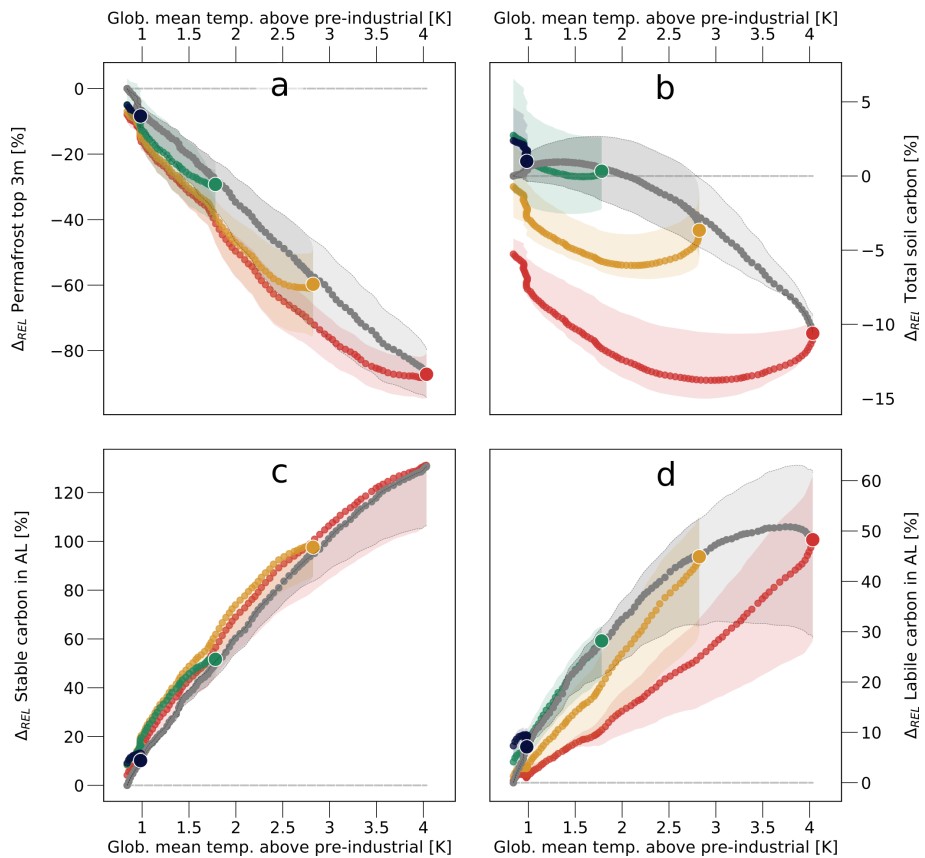

**Figure 7. Permafrost and soil carbon: a)** Changes in the volume of near-surface permafrost relative to the permafrost volume at the beginning of the 21st century **b)** Same as a but for the relative change in total soil carbon. **c)** Same as a but for the stable carbon within the active layer. **d)** Same as a but for the labile carbon within the active layer. The figure is representative of those areas that were affected by near-surface permafrost in the year 2000. Colours and symbols have the same meaning as in Fig. 6.

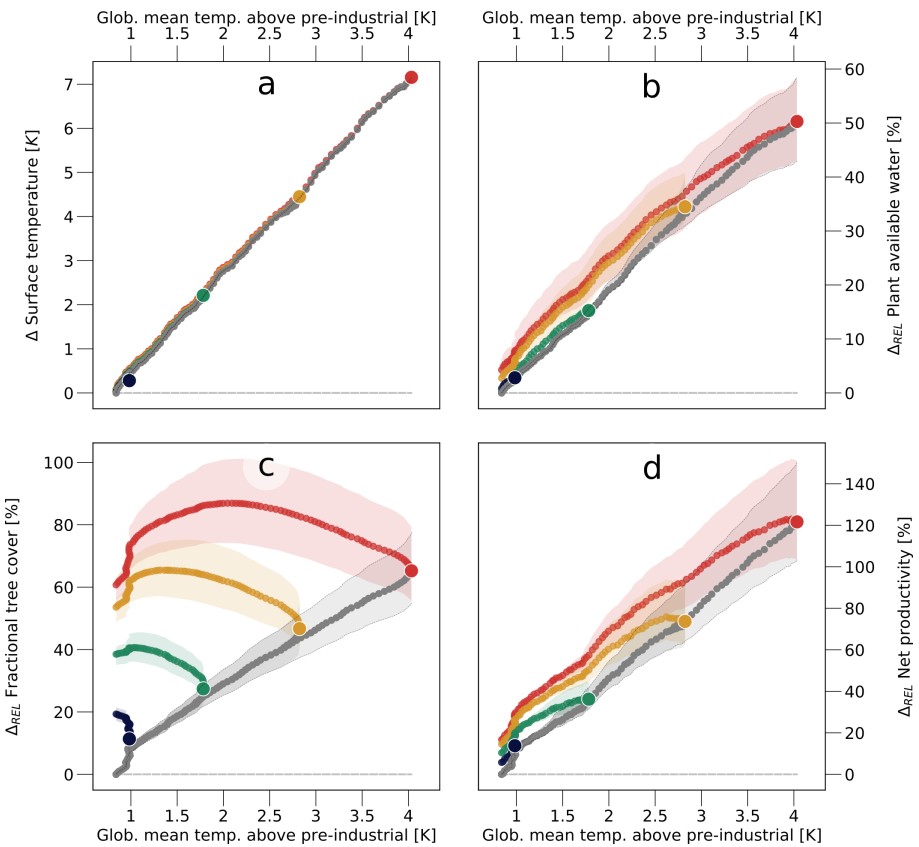

**Figure 8. Drivers of soil carbon inputs: a)** Change in mean surface temperature. Relative change in: **b)** Plant available water. **c)** Tree cover. **d)** Net primary productivity. All sub-figures pertain to regions with permafrost-affected soils. Colours and symbols have the same meaning as in Fig. 6.

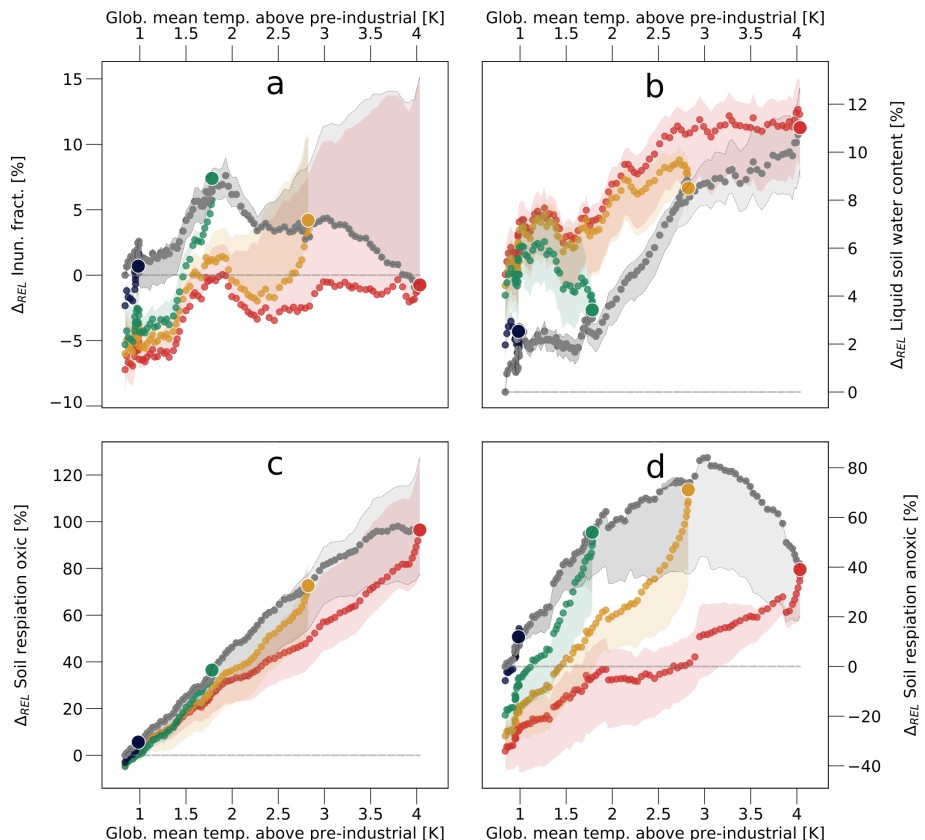

**Figure 9. Soil CH$_4$ and CO$_2$ production in seasonally inundated areas:** Relative change in: **a)** Extent of inundated areas. **b)** Liquid water content of the soil. **c)** Oxic soil respiration. **d)** Soil methane production. Sub-figure a shows the simple spatial average over permafrost-affected grid boxes, while sub-figures b-d include a weighting by the (annual-mean) wetland fraction. Colours and symbols have the same meaning as in Fig. 6.

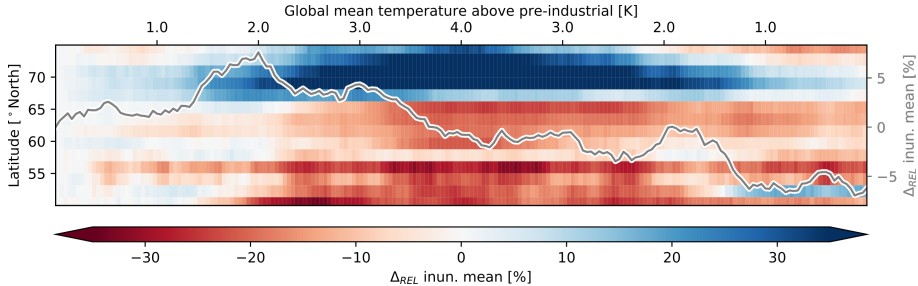

**Figure 10. Wetland area in permafrost-affected regions:** Relative change in the annual mean inundated area as a function of (global mean) temperature change and latitude (color-bar and left y-axis) and averaged over the permafrost-affected regions (grey line, right y-axis). Shown is the simulation with a temperature peak in 2100.

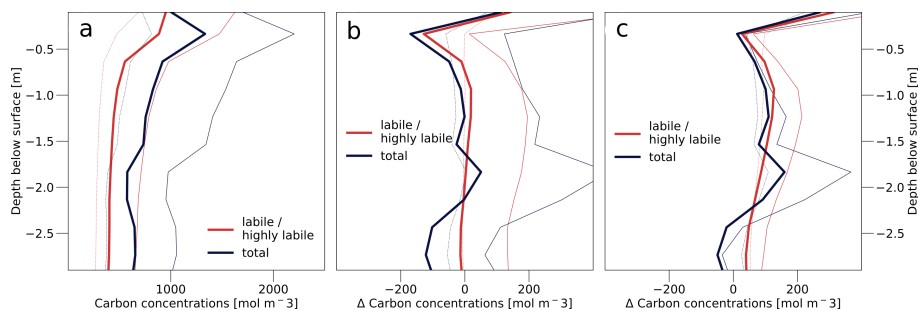

**Figure 11. Soil carbon profiles in seasonally inundated areas: a)** Carbon concentration as a function of soil depth in the year 2000 (1990 - 2010 average). Red lines show the concentration of labile and highly labile organic matter while blue lines represent the total carbon concentration. Thick solid lines show the ensemble mean, while dotted lines indicate the ensemble minimum and thin solid lines the ensemble maximum. **b)** Change in soil carbon concentrations between the forcing peak in the year 2100 (2090 - 2110 average) and the year 2000 (1990 - 2010 average). **c)** Same as b, but showing the differences between the years 2200 – when the forcing is fully reversed after a GHG peak in the year 2100 – and 2000. All sub-figures show the average over permafrost-affected grid boxes, weighted by their (annual-mean) wetland fraction.

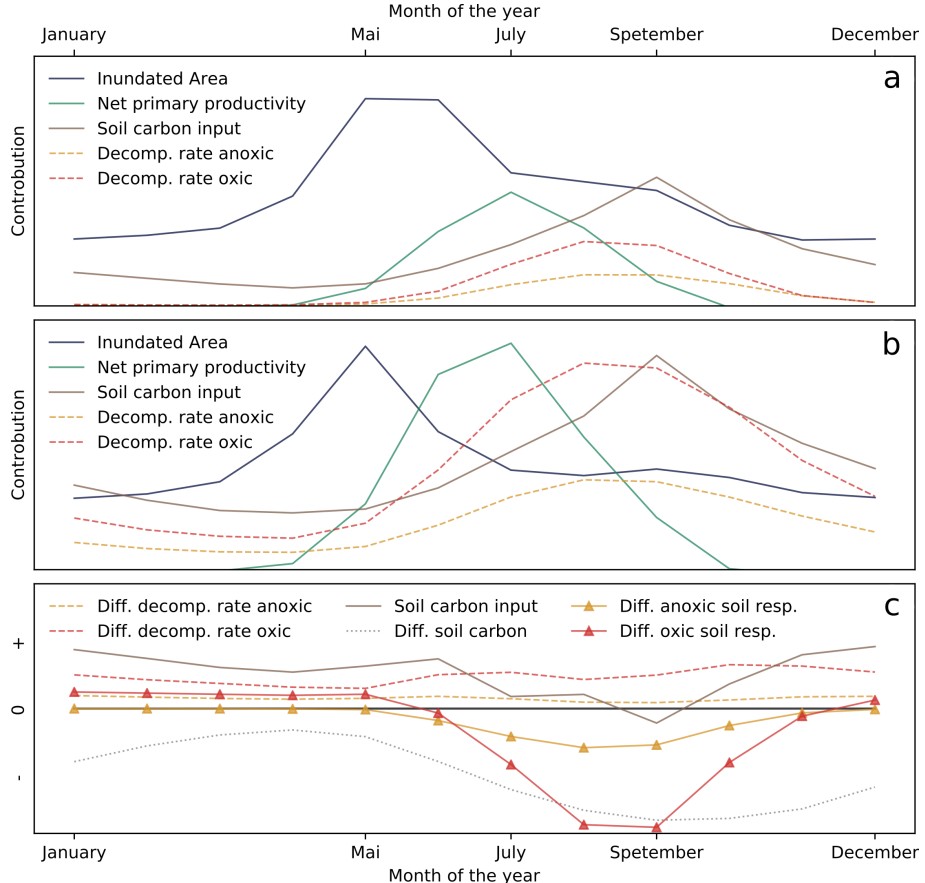

**Figure 12. Seasonal dynamics in inundated areas: a)** Annual cycle of inundated areas (blue lines), NPP (green), carbon input into the soil (brown), decomposition rates under anoxic conditions (yellow) and under oxic conditions (red) in permafrost-affected regions that feature a large wetland extent. Shown (qualitatively) are the seasonal dynamics that are representative for the year 2000, before the increase in atmospheric GHG concentrations. **b)** Same as a, but representative for a given forcing peak. **c)** (Qualitative) Differences in anoxic (dashed yellow lines) and oxic (red dashed lines) decomposition rates, carbon input into the soil (solid brown line), organic matter within the active layer (dotted grey line), anoxic (solid yellow line) and oxic soil respiration rates (solid red line) after and before a given forcing peak.

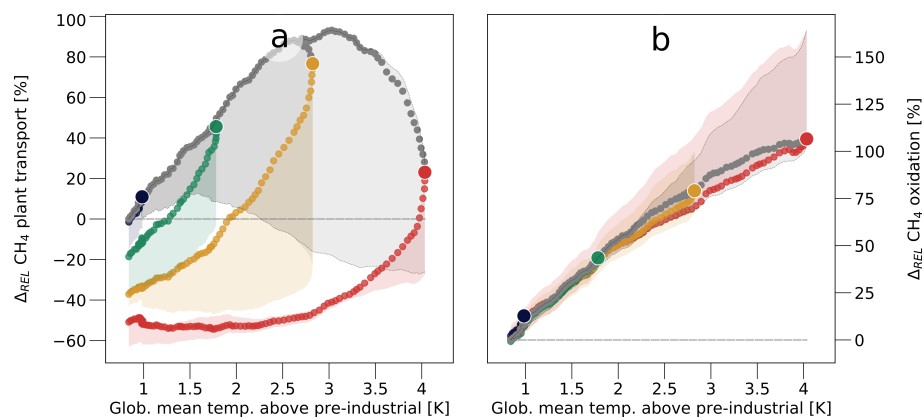

**Figure 13. Methane transport:** Relative change in: **a)** Amount of methane that is taken up by roots and emitted at the surface. **b)** Amount of methane that is oxidised within the soil. Colours and symbols have the same meaning as in Fig. 6.

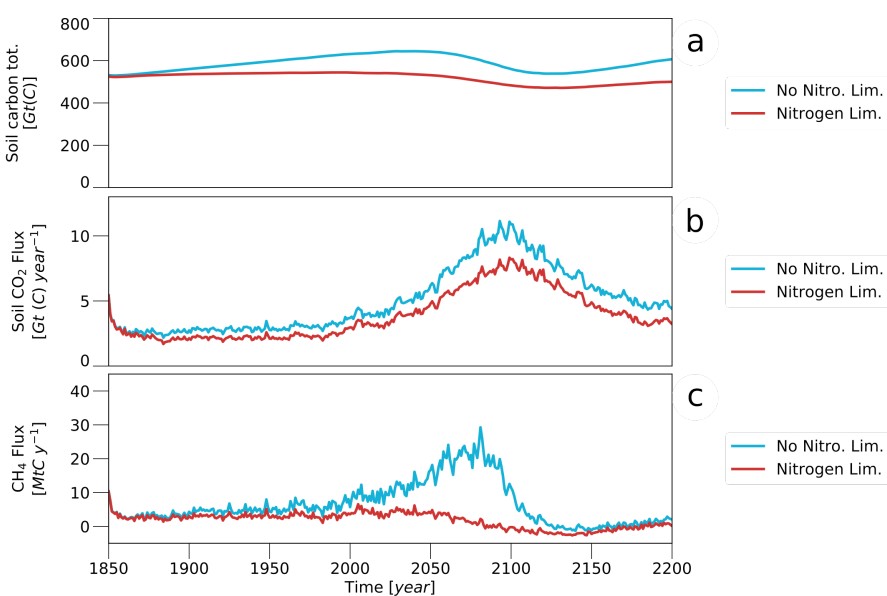

**Figure A1. Nitrogen limitations: a)** Soil organic matter in permafrost-affected regions simulated with (red line) and without (blue line) accounting for nitrogen limitations. **b)** Same as a but for the soil $CO_2$ fluxes. **c)** Same as a but for for soil methane emissions. Shown is the spinup period and the scenario with a temperature peak in the year 2100.

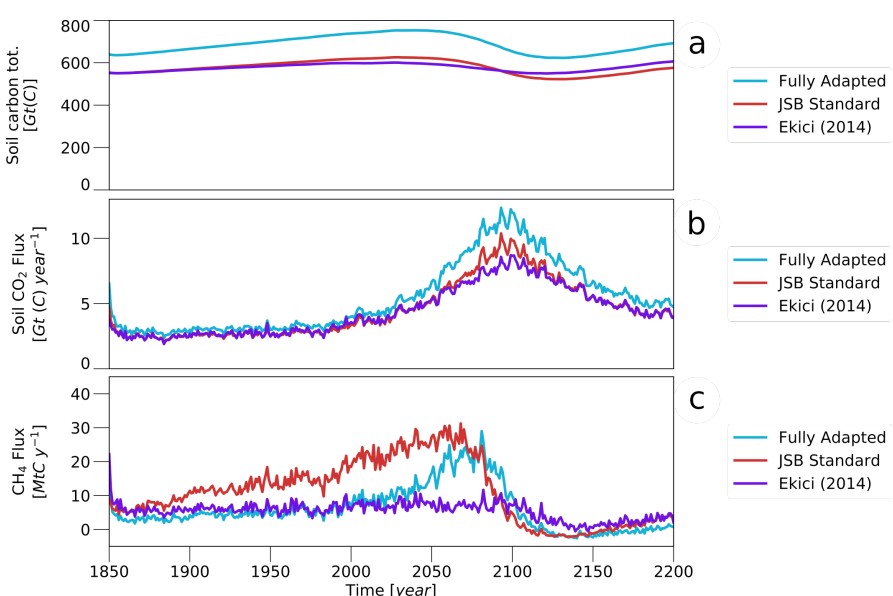

**Figure A2. Soil physics:** Same as Fig. A1 but showing differences with respect to key assumptions in the JSBACH soil-physics module. Light blue lines show simulations with the fully adapted model as described in section 2. Red lines show simulations that are closer to the standard JSBACH model, in that they assume vertically homogeneous organic matter properties, while infiltration at sub-zero temperatures is prevented and the supercooled water can move vertically through the soil and is available for microbial decomposition. Purple lines show simulations that account for the impact of organic matter on the soil properties only at the top of the soil column, while the lower layers have the properties of mineral soil. The latter provides a model version that is close to the setup used by Ekici et al. (2014, 2015).

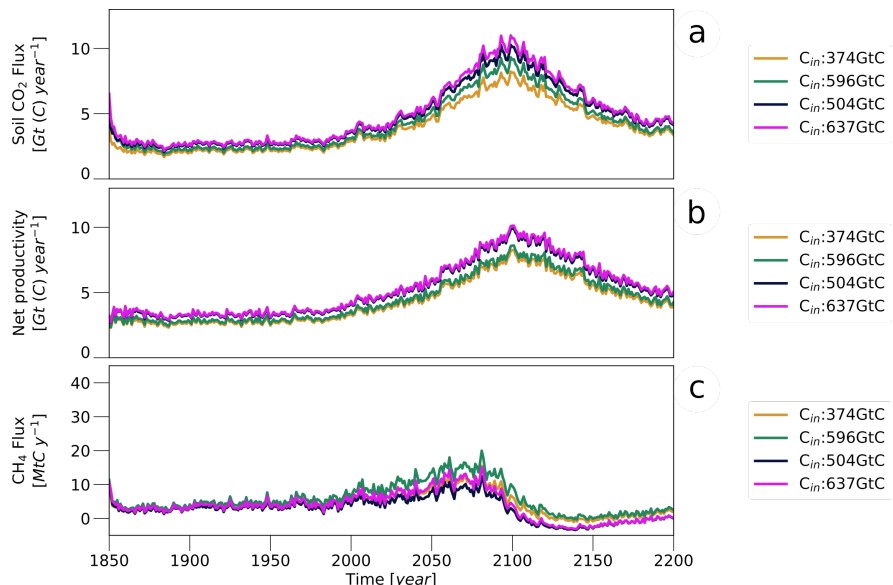

**Figure A3. Initial carbon pools: a)** Soil $CO_2$ fluxes in permafrost-affected regions simulated with different initial soil carbon concentrations and soil depths. **b)** same as a but for net primary productivity. **c)** same as a but for methane emissions. Yellow lines show simulations that were initialized with the observed soil organic matter that can be contained in the soil when using standard soil depths; roughly 636 GtC in regions that in reality are affected by continuous, discontinuous and sporadic permafrost and 374 GtC in those regions in which the model simulations near-surface permafrost in the year 2000. Green lines show simulations in which the carbon pools where scaled to be closer to the observed observed soil organic matter – roughly 1000 GtC. Here, the simulations where initialized with 858 GtC in the observed and 596 GtC in the simulated permafrost regions. Dark blue lines show simulations that were initialized with the observed soil organic matter concentrations but with increased soil depths in the model; these simulations are initialized with 797 GtC in the observed and 504 GtC in the simulated permafrost regions. Magenta lines show simulations with both, increased soils depths and upscaled initial soil carbon pools; these simulations are initialized with 931 GtC in the observed and 637 GtC in the simulated permafrost regions.

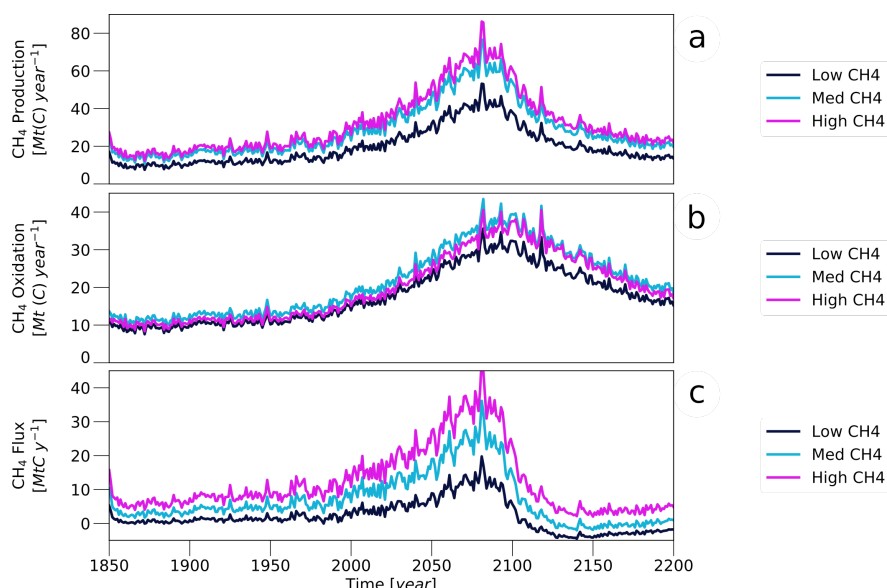

**Figure A4. Key parameters of the methane module: a)** Soil methane production in permafrost-affected regions simulated with different parameter combinations in the methane module that result in overall low (dark blue line), medium (light blue line) and high (magenta line) $CH_4$ emissions. The medium emission parameter combination was used for most of the simulations in the study and assumes a baseline fraction of $CH_4$ production of 0.4 and a Q10 factor of 1.5. For the low emission setup the baseline fraction of $CH_4$ was reduced to 0.35 and the Q10 factor was set 1.75. For the high emission simulations the baseline fraction of $CH_4$ was raised to 0.45 and the maximum oxidation velocities were reduced by 50 %.

**Table 1.** Overview over key variables in regions affected by near-surface permafrost. The first half of the table (2000,2025,2050,2075,2100) represents pools, states and fluxes during the increase in GHG concentrations according to the SSP5-8.5 scenario. The second half represents the post-peak (pp) pools, fluxes and states when the forcing has been fully reversed. 2050pp refers to the year 2050 after a GHG peak in the year 2025, 2100pp to a peak in 2050, 2150pp to the peak in 2075 and 2200pp to a peak in the year 2100. Black numbers indicate the ensemble mean and the grey number in brackets indicate the ensemble minimum and maximum. All variables represent simple spatial averages over the region affected by near-surface permafrost and averages over a 20-year period.

| Variable | Unit | 2000 | 2025 | 2050 | 2075 | 2100 | 2050pp | 2100pp | 2150pp | 2200pp |
|---|---|---|---|---|---|---|---|---|---|---|
| Soil carbon tot. | [GtC] | 606.7 (373.4/763.5) | 612.0 (371.7/779.2) | 608.3 (365.9/783.1) | 583.9 (352.7/753.8) | 542.4 (338.2/685.0) | 620.9 (372.8/798.1) | 622.8 (369.0/812.7) | 601.7 (362.8/778.8) | 574.1 (357.4/722.9) |
| Highly labile carbon in AL | [GtC] | 34.1 (20.8/43.5) | 37.5 (21.9/49.2) | 42.7 (23.0/59.3) | 45.9 (23.2/68.7) | 40.2 (23.5/56.6) | 36.0 (21.2/47.4) | 36.1 (21.6/48.2) | 35.3 (22.6/45.0) | 35.5 (24.0/43.1) |
| Labile carbon in AL | [GtC] | 87.0 (61.6/102.2) | 97.8 (68.0/115.9) | 113.9 (75.9/137.3) | 128.5 (81.3/159.2) | 129.0 (79.2/165.1) | 94.1 (65.2/111.9) | 91.3 (63.2/109.6) | 88.7 (61.9/105.8) | 87.9 (63.7/102.3) |
| Stable carbon in AL | [GtC] | 124.1 (89.0/152.5) | 148.6 (105.2/183.6) | 194.5 (134.3/240.6) | 251.6 (168.6/304.7) | 286.2 (183.6/342.8) | 137.1 (97.2/168.8) | 136.4 (98.2/169.7) | 136.3 (98.0/168.3) | 131.1 (93.5/160.5) |
| Net carbon flux | [GtC $year^{-1}$] | -0.34 (-0.81/0.08) | -0.32 (-0.81/0.11) | 0.15 (-0.21/0.41) | 1.03 (0.29/1.68) | 1.11 (0.14/2.24) | -0.36 (-0.78/0.01) | -0.47 (-0.87/-0.07) | -0.47 (-0.88/-0.09) | -0.44 (-0.84/-0.04) |
| Net primary productivity | [GtC $year^{-1}$] | 3.91 (2.49/5.01) | 4.58 (2.96/5.8) | 5.59 (3.75/6.94) | 7.17 (5.04/8.61) | 8.7 (6.2/10.2) | 4.15 (2.68/5.27) | 4.33 (2.73/5.56) | 4.5 (2.8/5.8) | 4.59 (2.84/5.92) |
| Soil $CO_2$ flux | [GtC $year^{-1}$] | 3.39 (2.4/3.97) | 4.02 (2.82/4.73) | 5.43 (3.72/6.43) | 7.75 (5.06/9.72) | 9.2 (6.0/11.7) | 3.59 (2.51/4.28) | 3.67 (2.51/4.45) | 3.83 (2.6/4.66) | 3.95 (2.7/4.81) |
| Soil $CH_4$ flux | [MtC $year^{-1}$] | 6.9 (2.8/19.3) | 8.2 (2.5/22.4) | 11.1 (1.4/26.7) | 13.4 (-1.2/35.7) | 6.0 (-2.2/21.9) | 6.6 (1.6/20.3) | 4.7 (0.3/13.7) | 2.7 (-0.2/10.3) | 1.28 (-2.17/4.34) |
| Soil methane production | [MtC $year^{-1}$] | 23.0 (16.8/38.1) | 27.1 (18.6/43.4) | 35.5 (21.7/51.7) | 43.7 (23.7/68.5) | 39.3 (25.2/59.6) | 23.3 (16.1/39.0) | 20.9 (14.5/30.6) | 19.1 (13.9/28.1) | 17.7 (13.2/23.9) |
| $CH_4$ plant transport | [MtC $year^{-1}$] | 9.4 (5.1/21.7) | 11.2 (5.3/25.2) | 15.1 (5.5/30.4) | 18.2 (4.2/39.9) | 11.6 (3.7/27.5) | 9.4 (4.4/22.6) | 7.8 (3.3/16.7) | 6.0 (3.0/13.4) | 4.69 (2.33/8.13) |
| $CH_4$ oxidation | [MtC $year^{-1}$] | 16.1 (13.3/18.9) | 18.9 (15.4/21.3) | 24.4 (19.2/28.4) | 30.4 (23.4/39.8) | 33.3 (26.8/49.8) | 16.7 (13.6/19.3) | 16.2 (13.0/19.7) | 16.3 (13.0/20.2) | 16.4 (12.9/20.5) |
| Soil temp. at 1m | [K] | 267.5 (267.0/268.0) | 268.4 (267.8/268.9) | 270.2 (269.6/270.8) | 272.6 (272.0/273.1) | 274.5 (274.1/274.8) | 267.9 (267.4/268.4) | 268.0 (267.4/268.5) | 268.1 (267.6/268.5) | 268.2 (267.7/268.6) |
| Permafrost in top 3m | [m] | 2.02 (1.84/2.16) | 1.79 (1.53/1.97) | 1.38 (1.05/1.62) | 0.74 (0.4/1.05) | 0.26 (0.1/0.43) | 1.91 (1.69/2.06) | 1.91 (1.67/2.2) | 1.87 (1.67/2.03) | 1.85 (1.63/2.02) |
| Inundated fract. (mean) | [%] | 4.51 (4.05/6.05) | 4.57 (4.12/6.01) | 4.85 (4.41/6.42) | 4.7 (4.29/6.74) | 4.47 (4.04/6.95) | 4.45 (4.01/6.02) | 4.32 (3.88/5.94) | 4.28 (3.8/5.88) | 4.23 (3.73/5.85) |

## Appendix A: Uncertainty

### A1 Nitrogen limitations

One of the main sources of uncertainty in the present experimental setup stems from the representation of nitrogen limitations and their impact on the carbon cycle in permafrost-affected regions. In the fully adapted model, the nitrogen limitations lower the vegetated fraction by up to 20%, while primary productivity and soil respiration rates are reduced by up to 35%. In contrast, these limitations merely amount to a few percent in the standard model (not shown). Consequently, the nitrogen availability strongly inhibits the soil organic matter build up during the spinup phase and the soil carbon concentrations remain close to the respective initial values (Fig. A1a). In contrast, when nitrogen limitations are neglected, the initial carbon pools are far from being in equilibrium with the simulated climate and the amount of soil organic matter increases by around 100 GtC during the spinup period. The reduced availability of decomposable matter that results from the nitrogen limitations lowers the $CO_2$ emissions from permafrost-affected soils by around 25%, corresponding to reduction of up to 2.5 GtC year$^{-1}$ at the end of the 21st century (Fig. A1b).

In our model, the availability of nitrogen is maily determined by the deposition flux and soil nitrogen mineralization on the one hand and by denitrification and the amount of nitrogen that is leached from the soils on the other hand. The latter depends on the amount of water that infiltrates and subsequently drains from the soil, making the nitrogen limitations particularly strong in regions that receive large amounts of precipitation and feature porous soils which facilitate infiltration at the surface. Previous studies found that the increased nitrogen mineralization stemming from the decomposition of formerly frozen soil organic matter could largely offset nitrogen limitations in a warming scenario (Koven et al., 2015) and our results also show that the effect of nitrogen limitations on productivity is reduced from roughly 35% during the spinup period to about 20% at the temperature peak. However, the increase in nitrogen availability is smaller in regions that are characterized by high infiltration and drainage rates, where a larger part of the mineralized and deposited nitrogen is leached from the soils during the spring snowmelt season. As these are also the areas that feature large wetland fractions, nitrogen limitations are particularly strong in methane-producing regions. Consequently, the effect of nitrogen limitations on the $CH_4$ fluxes is much more pronounced than the effect on the soil $CO_2$ emissions (Fig. A1c). Here the difference between the simulations with and without accounting for nitrogen limitations suggests that the behaviour of the methane fluxes under a future warming could largely be determined by nutrient availability. While the methane production in saturated soils increases with rising temperatures, even when accounting for nitrogen limitations (not shown), the oxidation rates increase similarly with the rise in atmospheric $CH_4$ concentrations. As a result, there is only a minor increase in the soil net $CH_4$ flux, and the emissions decline as early as the year 2050. When neglecting nitrogen limitations, the productivity in methane-producing regions is higher, leading to more soil organic matter, which increases the soil methane production, roughly doubling the net $CH_4$ emissions between the years 2000 and 2075. However, the net fluxes decrease before the temperature peak is reached in 2100 and the hysteresis-like behaviour is particularly pronounced in simulations that do not account for nitrogen limitations.

While the nitrogen limitations in the high latitudes are much larger in the adapted model setup than in the standard one, they are much closer to other model-based estimates. For example, Koven et al. (2015) calculated plant type specific (NPP) limitation factors of up to 30% for arctic vegetation. Thus, while the uncertainty with respect to the actual magnitude of the high latitude nitrogen limitations is substantial, there is some indication that our simulations that account for these limitations are much closer to reality than the simulations in which they are neglected.

## A2   Soil physics

As stated in Sec. 2 a number of modifications to JSBACH was required for this study and reversing some of the key aspects changes the simulations substantially. When preventing infiltration at sub-zero temperatures while allowing the supercooled water to move vertically through and drain from the soil – as is the case in the standard model –, the soils are substantially drier during spring and summer. This reduces the cooling due to evapotranspiration, leading to warmer soils and a reduction in the spatial extent of areas that are affected by near-surface permafrost. With a smaller permafrost area the initial amount of carbon stored in permafrost-soils is substantially lower than in simulations with the fully adapted model (Fig. A2a). In addition, the reduced water-availability inhibits the vegetation productivity (not shown), resulting in a reduced carbon build-up during the spinup period and substantially lower $CO_2$ emissions during and after the temperature increase than in the simulation with the fully adapted model (Fig. A2b).

Limiting the effect of soil organic matter on the soil properties to the first soil layer (as is the case in the setup of Ekici et al. (2014)) has a very similar effect because mineral soils have a lower water holding capacity and a larger heat conductivity than organic matter, leading to a lower primary productivity, reduced evapotranspiration rates and higher below-ground temperatures during summer. Furthermore, the dryer soils reduce the spatial extent of saturated soils and there is almost no increase in methane emissions with rising temperatures (Fig. A2c). In this respect the setup is very different not only from the fully adapted model, but also from the setup that is closer to the standard model. In the latter, the assumption that the supercooled water can be used in the process of decomposition leads to a non-negligible methane production at temperatures below the freezing point. Consequently, the net emissions, especially during the spinup period, are substantially higher than in any other setup. Here, it should be noted that it is not necessarily impossible to have decomposition within the frozen fraction of the soil, as microbial activity is not limited to temperatures above the freezing threshold, and there is evidence of substantial cold-season emissions (Zona et al., 2015). However, these emissions are thought to mainly occur close to the zero-curtain period and a large methane production within the permafrost at temperature below 0°C contrasts with the general understanding that organic matter within frozen soils is essentially inert. Thus, the high methane emissions that are simulated when assuming the supercooled water to be available for decomposition cannot be considered to be a very likely feature of the high latitude carbon cycle.

## A3 Initial carbon pools

As small-scale spatial variability in soil depths cannot be represented by a coarse resolution model, it is impossible to initialize the simulations with the observed soil carbon concentrations directly and we used 2 approaches to bring the initial soil carbon pools closer to observation-based estimates (Sec. 2). To cover the resulting uncertainty-range, we used 4 sets of carbon pools to initialize our simulations that are based on: The observed organic matter concentrations located above the standard bedrock border of the model (Fig. A3, yellow line – 374GtC), up-scaling the observed organic matter concentrations located above the standard bedrock border of the model (green line – 596GtC), observed organic matter concentrations located within the soil for a deep-soil-setup (blue line – 504GtC) and up-scaling the observed organic matter concentrations located within the soil for a deep-soil-setup (magenta line – 637GtC).

During the spinup phase the initial carbon pools only have a minor impact on the soil $CO_2$ emissions as the largest fraction of the organic matter is located within the simulated permafrost (Fig. A3a). In this phase, the existing differences in the simulated emissions are mainly a result of the different soil depths. The deeper soils increase the plant available water, reducing limitations on productivity which in turn increases the availability of decomposable material (Fig. A3b). Thus, there are only minor differences in the $CO_2$ fluxes between simulations that use the same soil depths (yellow and green lines; blue and magenta lines) and only after the year 2050 do the differences in initial soil organic matter affect the emissions, leading to notable differences between all the simulations. When the permafrost has largely reestablished with decreasing temperatures (roughly following the year 2150), the differences between the simulated $CO_2$ fluxes are again mainly caused by the differing NPP rates resulting from the differences in assumed soil depths. Thus, overall the simulated $CO_2$ fluxes are more sensitive to the soil depths than to the initial carbon pools – at least for the range that was investigated in our study – which is in agreement with our finding that the high latitude soil $CO_2$ emissions are to a large extent driven by the input of fresh organic matter, hence by NPP.

The simulated $CH_4$ fluxes are even more sensitive to the soil depths of the model. With comparable precipitation and evpotranspiration rates, the deeper soils are predominantly less saturated and the extent of inundated areas in which methane is produced is smaller (not shown). Consequently, the methane emissions in the simulations with the deeper soils are around 30% lower than in the simulations with the shallower soils, despite the larger initial soil carbon pools and the higher carbon input (Fig. A3c, compare yellow line and blue line). Furthermore, up-scaling the initial carbon pools to better match the observations increases the organic matter concentrations especially in regions that have deeper soils and already feature large carbon pools. In case of the standard soil depths, these are also the regions – especially around the Hudson bay area and the western Siberian lowlands – that exhibit large wetland fractions, raising the simulated $CH_4$ emissions substantially. For the deep-soil setup, this effect is less pronounced because the up-scaling is (to a larger extent) also applied to the organic matter located in dryer regions. Thus, the differences in the simulated $CH_4$ emissions between the deep and the standard soil setup is even more pronounced in case of the up-scaled initial carbon pools (Fig. A3c, compare green line and magenta line).

## A4   Methane module

The simulated $CH_4$ emissions are very sensitive to a number of key parameters in the methane module, most importantly the baseline fraction of $CH_4$, the Q10 factor and the assumed (maximum) oxidation velocities. For most of the simulations analysed in the present study (Fig. A4, light blue lines), the model uses a baseline fraction of $CH_4$ production of 0.4, a Q10 factor of 1.5 and a maximum oxidation rate of $1.25e^{-5}$ ($1.25e^{-6}$) mol m$^{-3}$ s$^{-1}$ in saturated (non-saturated) soils. With these parameter-settings the model simulates present-day wetland emissions of around 11 Tg(CH4) in the region between 60° and 90° North, which is in good agreement with other model- and inversion-based estimates (Saunois et al., 2020).

The values for these parameters are highly uncertain and decreasing the baseline fraction of $CH_4$ while increasing the Q10 factor by less than 20 % (Fig. A4, dark blue lines), lowers the methane production by about 30 %, resulting in close to (net-) zero emissions during the spinup phase and a methane uptake of the soil following the temperature peak in 2100. In contrast, increasing the baseline fraction of $CH_4$ by 13 % and decreasing the maximum oxidation rates by 50 %, raises the net emissions prior to the temperature peak by a factor of between 1.5 and 2 (Fig. A4, magenta lines).

However, while the absolute values are extremely different, the (relative) spatial patterns are very similar between the setups, as these are largely determined by the simulated wetland areas. Even more importantly, all simulations show a consistent response to increasing and decreasing temperatures (and atmospheric GHG concentrations) with substantially lower $CH_4$ fluxes after the temperature peak than before. Thus, the hysteresis-like behaviour of the methane emissions is a robust feature that does not depend on the parameter-settings of the methane module.