# Peer review of "Diverging responses of high latitude CO2 and CH4 emissions in idealized climate change scenarios"

_The Cryosphere, 2020_

## Referee Comment (RC1) · Alexey V. Eliseev (Referee) · 1 Sep 2020

As a whole, the paper is well written. The topic is timely in light of the ongoing attempts to understand and quantify the multi-centennial climate response to the continuing anthropogenic greenhouse gas emissions and their decline in future.

Thus, I vote for publishing this manuscript subject to addressing the following comments.

[Figure]

[Figure]

**Major comments**

The major comment to the paper is due to the lack of studying the regional pattern of the hysteresis–like phenomenon in the manuscript. Eliseev et al. (2014) found that the hysteretic response of permafrost extent is due to strong difference in thermophysical properties between the mineral soil and peat. I expect that this issue could be applicable to this manuscript as well.

In addition, there is a subtlety in term 'hysteresis'. In physics, this term is reserved for the response of a multi–stable system to change of an *externally imposed* governing parameter. This is different from the phenomenon studied in the present paper. Here, the hysteresis–like response is due to transient properties of the system under investigation — basically, because of difference in response time scales between different compartments (e.g., due to different thermal inertia between peat and mineral soil in (Eliseev et al., 2014)). This is highlighted by the fact that both variables forming the hysteresis curve (e.g., in Figs. 3-6 of the manuscript) are *internal* variables of the system. As a result, term 'transient hysteresis' was introduced by Eliseev et al. (2014). I suggest to discuss this issue in the paper under review as well.

**Minor comments**

- ll. 74 and 789: The correct year for Eliseev et al.'s paper is 2014.

- l. 110: '...very different properties ...' Very different for thermophysical or for hydrological processes?

- l. 137: $z$ in Eq. (1) lacks units. Otherwise, this equation is ambiguous.

- l. 163: '...the number of days per year in which surface temperatures crossed

...'. I guess, it should be 'the day of the year when surface temperature crossed ...'.

- l. 201: it should be 'anaerobic or aerobic'.

- l. 201: it should be 'its shape'.

- l. 265: the better spelling would be 'soil chemical composition'.

- l. 270: the better spelling would be 'soil pore space'.

- l. 343: '2' and '4' in chemical formulae should be subscripts.

- l. 537: I guess, one of two numbers is wrong, because $9$ $\mathrm{MtC\,yr^{-1}}$ is $12\,\mathrm{TgCH_4\,yr^{-1}}$.

- Fig. 2: This figure is difficult to read. I suggest to place ensemble means in the left column and draw the maps in the middle and right columns as *differences* from these ensemble means. In addition, phrases like 'Ensemble–minimum thaw depth (annual maximum) ...' in caption to this figure is quite difficult to understand for a general reader. I suggest to put the wording in form 'Ensemble minimum for annual maximum thaw depth ...' and so on.

---

## Referee Comment (RC2) · Gerhard Krinner (Referee) · 9 Sep 2020

de Vrese and coauthors present modeling study that investigates the response of the continental high latitude carbon cycle under "idealized" transient climate change with trend inversion at different points in time during the course of this century. The paper is in general very clear and addresses the important question of how the carbon cycle reacts under overshoot scenarios. It provides interesting into the complex interplay between the numerous processes and factors that would determine the trajectory of the climate system under overshoot scenarios, although it is clear that the current uncertainties (appropriately acknowledged in the paper) preclude firm predictions of

the evolution of continental high latitude CO2 and CH4 fluxes under such scenarios.

The paper is well structured and also well written, although, as far as I can judge (I'm not a native speaker either), the English could be improved in many places (for example: 1 - there are many commas that would be in place in German but not in English; 2 - the possessive case is often wrongly used [" 's " should be used only for person, not for things as far as I know]; 3 - hyphenation is probably used too often between two nouns, just to give a few examples of what I think are repeated errors).

I have some comments and suggestions that I hope might be useful to clarify some aspects of the paper.

- L. 52-54: Clarify that large-scale models actually represent the thaw depth and do not represent processes like thaw settling and thermokarst, which occur faster ("abrupt"); these are indeed local processes but why should their occurrence by widespread?

- L.72-76: Indeed there isn't much literature focusing on the behaviour of the Arctic continental ecosystems under overshoot scenarios. But there are several global studies, I think, from which information about the Arctic might be extracted. Maybe also check what the IPCC SROCC and SR1.5 say?

- L.117: Hard to understand what is done here with r_cin. Maybe a schematic could help?

- L.125-129: The vertical discretization is better described later. The short description here is frustrating because one misses some detail.

- L.275: parameterization of permafrost acting against drainage: please justify (e.g. by citing appropriate references)

- L.365: "likely scenario" - in principle, the IPCC scenarios have no likelihood attached. Maybe sufficient to say that SSP585 is not "business as usual" (see the comment by Hausfather and Peters, Nature 2020)

[Figure]

- L.371: Please specify which member of the historical ensemble was used (presumably the first?)

- L.455: Unclear whether the permafrost-affected area changes in time and between the pertubed physics ensemble members in terms of the analysis, or whether it is fixed. What is the impact on the results?

- L.470: CH4 emissions small. Specify that this is also the case in terms of forcing in your model. Aren't these CH4 emissions a bit low compared to current estimates?

- L.491: higher end of previous estimates: I have the impression that the near-surface permafrost extent in the MPI model has a very strong sensitivity to GSAT, compared to other models. Is this correct? If yes, what is the reason? Is the Arctic amplification particularly strong in this model or does the soil react very quickly and strongly?

- L.506-519: This tree fraction hysteresis is interesting and intriguing. Can you discuss this a bit more? What happens exactly? Why aren't these trees here in the first place? Is this realistic?

-L.520-531: Discussion a bit unclear. This got me really confused. Does this NPP increase lead to more litter? Is this increased litter fraction the reason for the emissions? Otherwise hard to see how there can be an emission increase without increasing soil carbon emissions. The carbon must go somewhere, and come from somewhere... Or does the vegetation carbon increase?

-L.660: At the end of this section, one wonders where all the sensitivity tests went. I have the impression that there could be made a better, clearer explicit use of the 40 members in terms of an assessment of the uncertainties.

-L.671: Soil methane oxidation increase: could refer to Oh et al. 2020 and discuss similarities & what is new

Some very minor specific suggestions:

- L.4: "drive the model" might be better than "force the model"

- Abstract, L.7: not only GHG decrease, but also reverse climate change is imposed on the land surface model

- L.32: Arctic temperature increase twice the global mean - it might be more appropriate to compare the Arctic contnintal temeprature change to the global continental average (but the numbers wouldn't be very different, probably)

- L.39: scenarios project a temperature increase between 3 and 8°C - it would be good to explicitly state that this uncertainty by 2100 comes to a very large degree from the diversity of the emission scenarios, not on the inter-model differences or internal variability

- L.43: timing of switch from sink to source highly uncertain - please provide some references here (maybe SROCC?)

- L.48: define what "near-surface permafrost" is.

- replace "arctic" or "artic" (found several times) by "Arctic"

- not sure "aerob" and "anaerob" are English words (should it read "(an)aerobic"?) - please check

- L.52: "permafrost-affectED soils"

- L.68: "the study's goal" -> "the goal of the study" (several such errors)

- L.72: Given that this refers to political temperature targets, it might be useful to use more post-Paris 2015 references here

- L.203: "be including" -> "by including"

- L.227 "Permafrost-physics" -> "Permafrost physics" (there are many more examples of what I suspect is wrong hyphenization is this text)

- L.335: "water tale" -> "water table"

- L.343: CO2 and CH4 -> 2 and 4 are index, please.

- L.343: Please consider providing the equation even though many people know what a Q10 is

- L.360: simulation period: CMIP6 historical period finishes in 2014, not 2015. Please check.

- L.382: "One key factor, determining..." - I think this is one example of a comma that shouldn't be there

- L.402, Eq. 25: "n_sim = n_c,soil *.... * n_c,CH4 = 40" (add "= 40") - would make things clearer

- L.690: "Le Quéré", not "Quéré"

- L.701: "not one model included an adequate representation..." - this might be a bit harsh. CCSM4, for example, probably isn't that far from being adequate, depending of course of what one thinks is adequate.
* * *

---

## Author Response (AR1)

Dear Editors,

As stated in our email from the 24th of November, we have changed our manuscript beyond what has been suggest in the review process. The reason being that we aim to use the simulations that were conducted for the present study also as a basis for a follow-up study in which we address the hysteresis that we found in permafrost-affected regions. In this follow-up study we aim to investigate the system's hysteretic behavior in the context of meeting the Paris agreements long-term target of maintaining temperatures at below 1.5C above pre-industrial levels. The latter is more of a real-world setting and, in order to have a seamless-transition between the two studies, we needed to revise some of the highly-idealized initial conditions that we used for the present investigation.

To be more specific, in the previous version of our manuscript, we initialized the carbon pools to be (close to being) in equilibrium with the simulated climate. However, in reality the soil carbon is far from being in equilibrium with the present-day climate and the pools are much smaller and still increase as the continental Arctic constitutes a substantial carbon sink. This discrepancy between real-world and our initial pools was very helpful because it made some of the effects of increasing temperatures more prominent in the idealized settings of the present study. However, it may not provide an ideal basis for discussing the real-world consequences of overshooting the Paris Agreement's temperature target.

Consequently, we modified the initial conditions of our experiment, changing to non-equilibrium carbon pools that are closer to the observations. Qualitatively, the findings of the study remain the same, but there are quantitative changes in our results.

- Most importantly, with the lower initial carbon pools the soil $CO_2$ emissions are reduced by about 10% -- corresponding to roughly -1 GtC/year in the year 2100.
- This reduces the ecosystem net carbon flux into the atmosphere and the permafrost-affected regions transition from carbon source to sink at a higher (global mean) temperature – now most simulations place the transition at a warming of around 1.75 K above pre-industrial levels, while in the previous version of the study this already happened at around 1.0 K above pre-industrial levels.
- The lower net emissions reduce the soil carbon loss associated with permafrost-degradation from 150 (+/- 50) GtC to about 60 (+/-20) GtC.

Additionally, we now include in the manuscript:

- A more detailed description of the procedure to obtain the initial soil carbon pools – including 2 new figures;
- A graphic comparison (1 figure) of simulations with the standard and our adapted JSBACH model, to demonstrate why the study required model-development;
- (We have reduced the overall number of ensemble members, allowing us to provide as an appendix) a short discussion of the key uncertainties – including 4 new figures.

Please find below the point-by-point address of the reviewers' concerns and the track changes between the present and previous manuscript versions.

With best regards,
Philipp de Vrese

**Response to comments of reviewer 1**

*Please note that, in the following point by point address, we repeat the reviewer's comments in red letters while **our response** is given in black letters. As we reran our simulations with different initial conditions, the manuscript changed after our initial response to the reviewers. All the post-reply updates are indicated in blue.*

**General**

As a whole, the paper is well written. The topic is timely in light of the ongoing attempts to understand and quantify the multi-centennial climate response to the continuing anthropogenic greenhouse gas emissions and their decline in future.Thus, I vote for publishing this manuscript subject to addressing the following comments.

**Major Comments**

The major comment to the paper is due to the lack of studying the regional pattern of the hysteresis–like phenomenon in the manuscript. Eliseev et al. (2014) found that the hysteretic response of permafrost extent is due to strong difference in thermo-physical properties between the mineral soil and peat. I expect that this issue could be applicable to this manuscript as well.

In addition, there is a subtlety in term 'hysteresis'. In physics, this term is reserved for the response of a multi–stable system to change of an externally imposed governing parameter. This is different from the phenomenon studied in the present paper. Here,the hysteresis–like response is due to transient properties of the system under investigation — basically, because of dif-ference in response time scales between different compartments (e.g., due to different thermal inertia between peat and mineral soil in (Eliseev et al., 2014)). This is highlighted by the fact that both variables forming the hysteresis curve (e.g., in Figs. 3-6 of the manuscript) are internal variables of the system. As a result, term 'transient hysteresis' was introduced by Eliseev et al. (2014).I suggest to discuss this issue in the paper under review as well.

We apologize for not giving the hysteresis-like behaviour the attention it deserved. However, we did conduct an extensive investigation into this feature, including several additional long-term simulations. Unfortunately, we did not see a way to adequately present the respective findings without substantially increasing the length of the manuscript. Thus, we decided to discuss the hysteretic behaviour and its spatial pattern in a separate study. Nonetheless, we are happy to provide a short summary addressing Dr. Eliseev's comments: We found three main factors that determine the dynamics; Most importantly,

the hysteretic behaviour is – as Eliseev et al. (2014) proposed – due to the large inertia of permafrost-affected soils. Here, the response time depends largely on the energy required for or released in the phase change of water, hence the signal indeed has a high spatial variability, with larger delays in regions with high soil water contents. Secondly, the timescales on which vegetation shifts occur and soil organic matter decomposes are much larger than the timescales of the imposed climate change. Thus the simulated vegetation and soil carbon pools simply lag behind the warming/cooling signal and the rise/decrease in CO2. The third factor is the change in the soil organic matter concentration that alters the hydrological/thermophysical soil properties. And even though the latter may initially not be the strongest of the factors it is highly important because it alters the boundary conditions under which physical and biophyiscal soil processes take place. Thus, the difference in soil carbon pools before and after the temperature peak has the potential to lead to an actual hysteresis – in the sense of multistability – rather than a transient hysteresis. To acknowledge that we can not estimate to which extent the hysteretic behaviour is transient in the present manuscript, we included the following statement at the end of the section describing the soil CO2 emissions: *It should be noted that the hysteretic behaviour arises partly because the characteristic timescales of the high-latitude carbon cycle – most importantly of vegetation shifts and the decomposition of soil organic matter – are larger than the timescales of the investigated climate change scenarios. In addition, high latitude soils have a large thermal inertia, especially due to the large amounts of energy required or released by the phase change of water within the ground. Thus, the simulated behaviour does not necessarily indicate the multistability of the system but may merely exhibit a transient hysteresis as described by Eliseev et al. (2014). However, the question whether the hysteresis is purely transient or indicative of multistability is beyond the scope of this study and the subject of an ongoing investigation.*

**Minor Comments**

- ll. 74 and 789: The correct year for Eliseev et al.'s paper is 2014.
  The reference was corrected to Eliseev et al. (2014).

- l. 110: '... very different properties... ' Very different for thermophysical or for hydrological processes?
  The text was changed accordingly.

- l. 137:zin Eq. (1) lacks units. Otherwise, this equation is ambiguous.
  Here, the units were added.

- l. 163: '... the number of days per year in which surface temperatures crossed... '. I guess, it should be 'the day of the year when surface temperature crossed... '.
  We actually use the "number of days" in the calculation of the vertical transport velocities. The idea behind this is that repeated thawing and refreezing leads to a more effective mixing of the soil properties within the active layer.

- l. 201: it should be 'anaerobic or aerobic'.
  The text was changed accordingly.

- l. 201: it should be 'its shape'.

  The text was changed to *"the shape of the litter"*.

- l. 265: the better spelling would be 'soil chemical composition'.

  The text was changed accordingly.

- l. 270: the better spelling would be 'soil pore space'.

  The text was changed accordingly.

- l. 343: '2' and '4' in chemical formulae should be subscripts.

  Spelling was changed to subscript numbers throughout the manuscript.

- l.537: I guess, one of two numbers is wrong, because 9 MtC yr-1 is 12 Tg CH4yr-1.

  Dr. Eliseev is absolutely correct and flux is indeed 12 Tg CH4yr-1.

  With the changes in the setup the fluxes change to 7 MtC yr-1 which corresponds to 9 Tg CH4yr-1.

- Fig. 2: This figure is difficult to read. I suggest to place ensemble means in the left column and draw the maps in the middle and right columns as differences from these ensemble means. In addition, phrases like 'Ensemble–minimum thaw depth (annual maximum)... ' in caption to this figure is quite difficult to under-stand for a general reader. I suggest to put the wording in form 'Ensemble mini-mum for annual maximum thaw depth... ' and so on.

  The figure was adapted accordingly (see below).

  Beyond Dr. Eliseev's suggestions we modified the figure to show the min, mean, and maximum inindated fraction with respect to the warmer period – May - October – instead of the whole year.

[Figure]

**Figure 1. Simulated permafrost, vegetated fraction and inundated areas in the year 2000: a)** Ensemble mean of the annual maximum thaw depth, corresponding to the year 2000 (1990 - 2010 mean). Grey areas indicate grid boxes in which the annual maximum temperatures throughout the top 3 m of the soil exceeded the melting point for more than 10 years in the period 1990 - 2010. These are considered to be unaffected by near-surface permafrost and are not taken into consideration in the study. **b)** Same as a but for the difference between ensemble minimum and mean. **c)** Same as a but for the difference between ensemble maximum and mean. **d)** Ensemble mean vegetated fraction. Note that this is the maximum grid box fraction that can be covered by vegetation, while the actual vegetated cover depends on the current state of the vegetation and can vary throughout the year. **e)** Same as d but for the difference between ensemble minimum and mean **f)** Same as d but for the difference between ensemble maximum and mean. **g)** Minimum inundated fraction during the summer months (May-October) for the year 2000. Shown is the ensemble mean. **h)** Same as g but for the May-October mean. **i)** Same as g but for the May-October maximum inundated fraction.

**References**

Eliseev, A. V., Demchenko, P. F., Arzhanov, M. M., and Mokhov, I. I.: Transient hysteresis of near-surface permafrost response to external forcing, Climate Dynamics, 42, 1203–1215, https://doi.org/10.1007/s00382-013-1672-5, https://doi.org/10.1007/s00382-013-1672-5, 2014.

**Response to comments of reviewer 2**

*Please note that, in the following point by point address, we repeat the reviewer's comments in red letters while **our response** is given in black letters. As we reran our simulations with different initial conditions, the manuscript changed after our initial response to the reviewers. All the post-reply updates are indicated in blue.*

**General**

de Vrese and coauthors present modeling study that investigates the response of the continental high latitude carbon cycle under "idealized" transient climate change with trend inversion at different points in time during the course of this century. The paper is in general very clear and addresses the important question of how the carbon cycle reacts under overshoot scenarios. It provides interesting into the complex interplay between the numerous processes and factors that would determine the trajectory of the climate system under overshoot scenarios, although it is clear that the current uncertainties (appropriately acknowledged in the paper) preclude firm predictions of he evolution of continental high latitude CO2 and CH4 fluxes under such scenarios.The paper is well structured and also well written, although, as far as I can judge(I'm not a native speaker either), the English could be improved in many places (for example: 1 - there are many commas that would be in place in German but not in English; 2 - the possessive case is often wrongly used [" 's " should be used only for person, not for things as far as I know]; 3 - hyphenation is probably used too often between two nouns, just to give a few examples of what I think are repeated errors).I have some comments and suggestions that I hope might be useful to clarify some aspects of the paper.

**Comments**

- L. 52-54: Clarify that large-scale models actually represent the thaw depth and do not represent processes like thaw settling and thermokarst, which occur faster ("abrupt"); these are indeed local processes but why should their occurrence by widespread?

  In hindsight, this sentence may have been misleading. By stating that the processes are locally confined we did not want to indicate whether or not they are widespread but merely that they are not captured by large-scale models – which makes it difficult to estimate how Arctic GHG-emissions will develop in the future. We hope to clarify this by changing the sentence to: *"While local observations indicate that the processes which affect the soil carbon emissions are often locally confined and act on very short timescales, large-scale models do not represent these small-scale processes. Thus, studies*

*relying on these models suggest that the increase in emissions is likely to occur gradually over a timescale of hundreds of years (Schuur et al., 2015)".*

– L.72-76: Indeed there isn't much literature focusing on the behaviour of the Arctic continental ecosystems under over-shoot scenarios. But there are several global studies, I think, from which information about the Arctic might be extracted. Maybe also check what the IPCC SROCC and SR1.5 say?

It is correct that there is a number of studies that look at overshoot scenarios on the global scale, including permafrost regions. However, – to the best of our knowledge – the models used for these studies predominantly lack the representation of relevant processes. Here, the MPI-ESM is an excellent example: The standard model does not include the organic matter stored in the perennially frozen parts of the ground and thus, misses the carbon release due to permafrost degradation. It also does not represent freezing and thawing of soil water and misjudges the timescale of the hysteresis. Thus, the issue with the focus is not only a question of spatial scales but also of model capabilities.

– L.117: Hard to understand what is done here with r_cin. Maybe a schematic could help?

Here we tried to clarify the use of r_cin by providing the respective equations: *"The present model version distinguishes between anoxic and oxic decomposition in the inundated and the non-inundated fractions of the grid box (see below) and the soil carbon pools need to be separated accordingly. Here, we do not simulate the respective pools explicitly. Instead we determine $r_{C_{in}}^{t_{end}}$, the ratio between the carbon concentrations in the inundated ($C_{in}^{t_{end}}$) and the non-inundated ($C_{dry}^{t_{end}}$) fractions, for each of the soil carbon pools after the decomposition is computed in timestep $t$.*

$$r_{C_{in}}^{t_{end}} = \frac{C_{in}^{t_{end}}}{C_{dry}^{t_{end}}} \tag{1}$$

*In the consecutive time step $t+1$, the soil carbon is distributed between inundated and non-inundated carbon pools according to $r_{C_{in}}^{t_{end}}$ before the decomposition is calculated.*

$$C_{in}^{t+1_{start}} = C_{tot}^{t+1_{start}} \left(1 + \frac{1}{r_{C_{in}}^{t_{end}}}\right)^{-1}, \tag{2}$$

$$C_{dry}^{t+1_{start}} = C_{tot}^{t+1_{start}} - C_{in}^{t+1_{start}}. \tag{3}$$

– L.125-129: The vertical discretization is better described later. The short description here is frustrating because one misses some detail.

We removed the description of the two grids entirely because the simulations that are being analyzed for the present study use the same vertical grid for physics and soil carbon. Hence, it is a technical detail of the scheme that is not relevant for the present manuscript and indeed somewhat out of place here.

– L.275: parameterization of permafrost acting against drainage: please justify (e.g. by citing appropriate references)

In general, we evaluated the parametrizations based on the simulated soil moisture profiles for selected grid boxes and

on the hydrographs of the larger Arctic rivers. With respect to soil moisture profiles in grid cells underlain by permafrost, we followed Swenson et al. (2012) who modified the soil hydrology in CLM in the presence of soil ice. Similar to CLM, the MPI-ESM initially featured extremely dry soils when accounting for freezing and melting of soil water, which is in poor agreement with observations often indicating high moisture levels within the active layer and the formation of a perched highly saturated zone on top of the perennially frozen soil layers (Swenson et al., 2012). Furthermore, the inhibition of drainage has to be seen in the context of the MPI-ESM's soil hydrology scheme. Even the standard 5-layer scheme assumes that water moves to the bedrock border before it drains (see Fig. 1 Hagemann and Stacke (2015)) and, conceptually, the lateral drainage from overlying layers is merely an additional flow pathway that facilitates the vertical transport towards the bedrock border. This pathway represents wider fissures, cracks etc. that are not explicitly represented in the model, but are assumed to be present in all grid boxes – given the coarse, standard spatial resolution of the model. Hence, the inhibition of the lateral drainage is conceptually a limit on the vertical transport. Finally, we still allow lateral drainage from any layer in the case that the underlying soil layers are fully saturated, which we did not mention in the manuscript before. We hope to clarify this by extending the paragraph to: *"Additionally, the standard model version assumes lateral drainage from all soil layers located above the bedrock. This drainage component is included to account for vertical channels, e.g., connected pathways in coarse material, cracks or crevices, that are assumed to be present in the large, heterogeneous grid cells at the standard resolution (1.9° × 1.9°). These efficiently transport the water deeper underground towards the border between soil and bedrock where it runs of as base flow. However, in the presence of permafrost, we assume these vertical channels to be predominantly blocked by ice and we allow lateral drainage only at the bedrock boundary or from those layers below which the soil is fully water saturated, i.e. at field capacity. These limitations on lateral drainage in combination with the inhibition of percolation for large ice contents facilitate high moisture levels within the active layer and the formation of a perched highly saturated zone on top of the perennially frozen soil layers, which are typical for permafrost regions (Swenson et al., 2012)"*.

– L.365: "likely scenario" - in principle, the IPCC scenarios have no likelihood attached. Maybe sufficient to say that SSP585 is not "business as usual" (see the comment by Hausfather and Peters, Nature 2020).

Dr. Krinner is right that there are no probabilities attached to the scenarios, thus "likely" is a somewhat problematic term – though Hausfather and Peters (2020) state that RCP8.5 was introduced as an "unlikely" scenario. To avoid the connotation of probabilities we now use "plausible scenario" in the manuscript.

– L.371: Please specify which member of the historical ensemble was used (presumably the first?)

We used the 10th ensemble member for our historical and scenario simulations and a corresponding statement is now included in the methods section. Here, the 10th member was chosen because it starts from the latest point of the pi-control simulation – after 449 years. However, this was an arbitrary choice as, to the best of our knowledge, the pi-control simulation is in an equilibrium state after the spin-up phase.

– L.455: Unclear whether the permafrost-affected area changes in time and between the pertubed physics ensemble members in terms of the analysis, or whether it is fixed.What is the impact on the results?

This is an important factor which we indeed did not discuss sufficiently in the manuscript. For the analysis, we used fixed (in time) masks that are based on the simulation-specific near-surface permafrost extent (top 3 m) of the year 2000. For each of the simulations, the variables were aggregated over this region and we analysed the resulting spatial averages/sums. The permafrost extent varies between the simulations by up to 20% – roughly between 13 million km$^2$ and 16 million km$^2$ – which is also noticeable in the results, e.g. the large spread in the soil carbon pools can partly be attributed to the differences in the permafrost area. To clarify our approach in the manuscript, we included the following paragraph following the description of the ensemble simulations: *"With respect to the results presented below, most of the analysis is performed based on aggregated values representative of the entire northern permafrost region – here defined as the areas that exhibit perennially frozen soils within the top 3 m of the soil column (Andresen et al., 2020). The extent of these areas is sensitive to the parameter values used in a specific setup and varies substantially between the simulations. For the analysis we do not define a shared permafrost mask, but aggregate the values based on the simulation-specific permafrost region. Furthermore, we base the analysis on the permafrost regions at the beginning of the 21st century – roughly between 13 million km$^2$ and 16 million km$^2$ – and do not adjust their extent to account for the changes in the near-surface permafrost. Nonetheless, we simply refer to the focus region as the permafrost region in the manuscript even though large fractions of the respective areas may not feature near-surface permafrost at the higher temperatures of the assumed warming scenarios."*

– L.470: CH4 emissions small. Specify that this is also the case in terms of forcing in your model. Aren't these CH4 emissions a bit low compared to current estimates?

As the first reviewer pointed out, the estimate of 9 Tg CH4yr-1 is a slip of the pen for which we apologize. The correct simulated emissions amount to 12 Tg CH4yr-1 which is in good agreement with recent estimates – e.g. the Global Methane Budget by Saunois et al. (2020) estimates 2 - 18 Tg CH4yr-1 for high latitude wetlands, with inversion models placing the emissions at around 13 Tg CH4yr-1 and land surface models at 9 Tg CH4yr-1. In the manuscript we corrected the value (9 -> 12 Tg CH4yr-1), included a reference to Saunois et al. (2020) and clarify that our study focuses on the natural emissions from to wetlands. To this end we included the following statement in the description of the methane-module:*"It should be noted that the model also simulates the CH$_4$ emissions from wildfires and termites. However, with the focus of this study on soil emissions, we neglect these fluxes in the detailed discussion of the methane emissions and exclusively report the fluxes from wetlands and inundated areas"* and clarify that *"At the beginning of the 21st century the simulated net CH$_4$ emissions from water saturated high latitude soils amount to roughly 9 Mt(C) year$^{-1}$ – or about 12 Tg(CH$_4$) year$^{-1}$, which is in good agreement with recent estimates of high latitude wetland emissions (Saunois et al., 2020)."*

Due to the changes in initial carbon pools and our selection of setups the simulated methane emissions change: *"At the beginning of the 21st century the simulated net CH$_4$ emissions from high latitude soils amount to roughly 7 MtC year$^{-1}$ – or about 9 Tg(CH$_4$) year$^{-1}$"*. Nonetheless, for the region north of 60N our module remains in the range of previous esitmates – with present day wetland emissions of about 11 Tg(CH$_4$) year$^{-1}$. *"With these parameter-settings the model*

*simulates present-day wetland emissions of around 11 Tg(CH4) in the region between 60° and 90° North, which is in*
*good agreement with other model- and inversion-based estimates (Saunois et al., 2020)".*

– L.491: higher end of previous estimates: I have the impression that the near-surface permafrost extent in the MPI model
has a very strong sensitivity to GSAT, compared to other models. Is this correct? If yes, what is the reason? Is the Arctic
amplification particularly strong in this model or does the soil react very quickly and strongly?

It is correct that in the high northern latitudes the model reacts strongly to the GSAT increase which, as Dr. Krinner
correctly speculated, is in large parts due to a strong Arctic amplification. While the MPIESM1.2 has a low climate
sensitivity, Arctic temperatures increase comparatively fast with rising $CO_2$s. Following RCP8.5/SSP5, the near surface
temperatures in the continental Arctic increase by about 10K (relative to 1960) by the year 2100, while other models
reach this threshold between 25 - 75 years later (see Fig. 2 in Andresen et al. (2020)). How the land surface reacts to
a temperature increase of 10K is also largely model dependent – e.g. while CLM loses only half of the near surface
permafrost (in terms of area), there is a number of models that appear to loose most of the near surface permafrost for a
10K warming. Here, the below-ground temperatures in the standard JSBACH model – at least the HR version – appear
to more sensitive to rising atmospheric temperatures than they are in other models (see Fig. 12 b, Burke et al. (2020)).
However, in our model version we have changed a number of important parameters – e.g. soil depth, number of soil
layers, soil properties depending on organic matter – and parametrizations – e.g. water in the soil freezes and melts –
which reduced this sensitivity substantially. In general, the reason why land surface models react so differently is, to the
best of our knowledge, still somewhat unclear because they are very similar in many aspects of the soil physics and there
doesn't appear to be a single characteristic that sets models with a strong reaction apart from those indicating a weaker
response to warming.

Here, we additionally included a figure showing a comparison of observed and simulated that depths, showing that our
adapted model version captures the present day thaw depths much better than the standard model version (Fig. 1a).

– L.506-519: This tree fraction hysteresis is interesting and intriguing. Can you discuss this a bit more? What happens
exactly? Why aren't these trees here in the first place?Is this realistic?

The hysteretic behaviour is indeed an interesting results of our simulations and we conducted an extensive analysis of
the underlying mechanisms. However, a detailed explanation, including the question whether the effects of a temporary
warming are fully reversible, is rather lengthy and beyond the scope of this study, especially as it requires several sets
of additional simulations. Unfortunately, we had to conclude that it is best to focus on the GHG emissions and discuss
the hysteresis in a separate study. However, we don't want to give the impression of avoiding Dr. Krinner's question
and a simplified answer is that the hysteretic behaviour stems partly from the representation of the vegetation dynamics
in JSBACH, where the transition from (predominantly) grass- to shrub lands to forests occurs on decadal timescales
and the simulated vegetation simply lags behind the warming/cooling signal and the rise/decrease in $CO_2$. Additionally,
the soil has a large inertia (due to the large amounts of energy required/released in the phase change of water), which

also affects the vegetation via the soil water availability and finally there is the effect of the organic matter on the soil thermal/ hydrological properties, which leads to the ground behaving differently after the soil carbon loss due to a temporary warming. The question whether the hysteresis is realistic is not easy to answer as there are no observations for comparable warming/cooling scenarios. Here, we can only say that the hysteresis is a highly plausible behaviour and a robust feature in all of the 40 simulations that we conducted. Furthermore, the dynamics of the near surface permafrost are consistent with the findings of Eliseev et al. (2014). Thus, we trust the tree cover hysteresis to be realistic to the extent to which we trust vegetation models in general.

– L.520-531: Discussion a bit unclear. This got me really confused. Does this NPP increase lead to more litter? Is this increased litter fraction the reason for the emissions?Otherwise hard to see how there can be an emission increase without increasing soil carbon emissions. The carbon must go somewhere, and come from somewhere... Or does the vegetation carbon increase?

Dr. Krinner is correct that the NPP dependency of the soil $CO_2$ fluxes arises mainly from above and below ground litter, but the model also includes root exudates, fires and windbreak. Here we specified: *"Another important driver of the soil $CO_2$ fluxes is the carbon input into the soil – consisting of litter, root exudates but also damaged and burnt vegetation – which is largely dependent upon the net primary productivity (NPP)."*

– L.660: At the end of this section, one wonders where all the sensitivity tests went. I have the impression that there could be made a better, clearer explicit use of the 40members in terms of an assessment of the uncertainties.

We agree that the ensemble spread is not an ideal way to deal with the uncertainties and we actually went a lot further in the respective analysis. Unfortunately, we do not see a way to integrate this analysis in the present study without drastically increasing the amount of text, especially as this requires a much more detailed description, not only of the soil hydrology/energy schemes of the standard model but also of those of the model version described in Ekici et al. (2014). Furthermore, many of the insights gained by analysing the ensemble (and additional sensitivity experiments) may also not be of great interest to the modelling community as they pertain to parametrizations/feedbacks that are very specific to JSBACH. Hence, we would prefer to show the ensemble spread, to demonstrate the large uncertainties even for a single land surface model with a prescribed atmospheric forcing, but without going into details with respect to their origin.

When rerunning our simulations with the new initial soil carbon pools, we greatly reduced the number of simulations – from 40 to 20 – which made it feasible to describe at least the main causes of uncertainty which we now present in the appendix.

– L.671: Soil methane oxidation increase: could refer to Oh et al. 2020 and discuss similarities & what is new.

In general, the oxidation rates in the methane module can be scaled for the wet- and dry grid box fraction separately, in principal allowing to distinguish between high(dry)- and low(wet)-affinity methanotrophs. Here, we tuned the parameters with the help of atmospheric inversions (performed by the MPI for Biogeochemistry) for present day conditions, managing to capture a reasonable methane uptake by dry upland soils. However, the parameter values may not be ideal

for future conditions if the temperature dependencies are substantially different between high- and low-affinity methanotrophs as indicated by Oh et al. (2020). This could mean that the future net methane emissions simulated by JSBACH could be an overestimation, which we acknowledge in the conclusion section: *"Thus, our results indicate that the soil methane fluxes in permafrost-affected regions do not constitute an important contributor to the climate-carbon feedback. Here, it should be noted that the net emissions could be even lower as a recent study has indicated that the methane uptake in dry soils could be severely underestimated due to the omission of recently identified high-affinity methanotrophs (Oh et al., 2020), especially under future climatic conditions".*

**Minor comments**

- L.4: "drive the model" might be better than "force the model"- Abstract,
  The text was changed accordingly.

- L.7: not only GHG decrease, but also reverse climate change is imposed on the land surface model
  Here, we specified that the entire climate forcing is reversed to the initial levels: *"The peaks are followed by a decrease in atmospheric GHGs that returns the concentrations to the levels at the beginning of the 21st century, reversing the imposed climate change".*

- L.32: Arctic temperature increase twice the global mean - it might be more appropriate to compare the Arctic continental temperature change to the global continental average(but the numbers wouldn't be very different, probably)
  Dr. Krinner is correct that for our study the terrestrial temperatures are more relevant. However, we did not change the manuscript because – as he correctly speculated – the numbers do not change fundamentally, while most studies discuss Arctic amplification without distinguishing between continental temperatures and sst.

- L.39: scenarios project a temperature increase between 3 and 8°C - it would be good to explicitly state that this uncertainty by 2100 comes to a very large degree from the diversity of the emission scenarios, not on the inter-model differences or internal variability
  We specified that the large spread is the result of the different GHG emissions assumed by the scenarios: *Depending on the assumed greenhouse gas (GHG) emissions, climate change scenarios project the Arctic temperatures to increase by between 3 K and 8 K until the end of the 21st century (Stocker et al., 2013)."*

- L.43: timing of switch from sink to source highly uncertain - please provide some references here (maybe SROCC?)
  In modelling studies, the timing of the sink-to-source switch appears to be highly model and scenario dependent, already providing a large degree of uncertainty. But while models predominantly place the switch from source to sink into the second half of the century, even for RCP8.5, a recent observation based study by Natali et al. (2019) indicated that winter emissions could be substantially larger than previously thought and that the continental Arctic may already be a net source of atmospheric CO2. In the manuscript we have complemented the references provided to now include Schuur et al. (2008); Schaefer et al. (2011); Koven et al. (2015); McGuire et al. (2018); Parazoo et al. (2018); Natali et al. (2019).

- L.48: define what "near-surface permafrost" is.- replace "arctic" or "artic" (found several times) by "Arctic"- not sure "aerob" and "anaerob" are English words (should it read "(an)aerobic"?) -please check

  Here we added that "near-surface" often refers to the top 3 m and all instances of "arctic" and "artic" were changed to "Arctic" while "(an)aerob" was replaced by "(an)aerobic".

- L.52: "permafrost-affectED soils"

  The text was corrected.

- L.68: "the study's goal" -> "the goal of the study" (several such errors)

  The manuscript was updated accordingly and, in general, we limit the use of the possessive to living things.

- L.72: Given that this refers to political temperature targets, it might be useful to use more post-Paris 2015 references here

  Here, we added Geden and Löschel (2017); Ricke et al. (2017); Rogelj et al. (2015, 2018)

- L.203: "be including" -> "by including"

  The text was corrected.

- L.227 "Permafrost-physics" -> "Permafrost physics" (there are many more examples of what I suspect is wrong hyphenization is this text)

  Here, we corrected "Permafrost-physics" -> "Permafrost physics". In general, we double(-)checked and reduced our use of hyphens.

- L.335: "water tale" -> "water table"

  The text was corrected.

- L.343: CO2 and CH4 -> 2 and 4 are index, please.

  Spelling was changed to subscript numbers throughout the manuscript.

- L.343: Please consider providing the equation even though many people know what a Q10 is

  Here, we included the equation for the production of CH4: *Partitioning of the anaerobic decomposition product ($R_{anox}$) into $CO_2$ and $CH_4$ ($P_{CH_4}$) is temperature-dependent, with a baseline fraction of $CH_4$ production $f_{CH_4}$ = 0.35 and a Q10 factor for $f_{CH_4}$ of Q10 = 1.8 – with a reference temperature ($T_{ref}$) of 295K.*
  $$P_{CH_4} = R_{anox} \cdot f_{CH_4} \cdot Q_{10}^{\frac{T_z - T_{ref}}{10}}$$"

  With the reduced initial soil carbon pools we re-tuned the methane modele to better match present day observations, resulting in the following paramter-setting: *"Partitioning of the anaerobic decomposition product ($R_{anox}$) into $CO_2$ and $CH_4$ ($P_{CH_4}$) is temperature-dependent, with a baseline fraction of $CH_4$ production $f_{CH_4}$ = 0.4 and a Q10 factor for $f_{CH_4}$ of Q10 = 1.5, with a reference temperature ($T_{ref}$) of 295K."*

- L.360: simulation period: CMIP6 historical period finishes in 2014, not 2015. Please check.

  Dr. Krinner is absolutely correct and the manuscript was changed accordingly.

- L.382: "One key factor, determining..." - I think this is one example of a comma that shouldn't be there

  We apologize for our (German) approach to punctuation and grammar in general, which we tried to correct throughout the manuscript.

- L.402, Eq. 25: "n_sim = n_c,soil *.... * n_c,CH4 = 40" (add "= 40") - would make things clearer

  The equation was updated accordingly.

  Here, we reduced the number of simulations and the respective text changend accordingly, but we followed Dr. Krinner's suggestion also in the new manuscript version.

- L.690: "Le Quéré", not "Quéré"

  The reference was updated accordingly (Le Quéré et al., 2018).

- L.701: "not one model included an adequate representation..." - this might be a bit harsh. CCSM4, for example, probably isn't that far from being adequate, depending of course of what one thinks is adequate.

  To the best of our knowledge, even CESM2 still had some problems with the high-latitude carbon stocks in the CMIP6 simulations (Danabasoglu et al. (2020); p. 26; reported a error in the spin-up phase). However – given the limitations of the CMIP6 version of the MPI-ESM – we should certainly not be the ones to judge what is adequate or not. Thus, we changed the respective formulation, leaving some room for a small number of models to have met the criteria: *"...,  but hardly any of the models that participated in CMIP6 included an adequate representation of the soil physics in high latitudes, while simulating (interactive) vegetation dynamics as well as the carbon and nitrogen cycle".*

[revised manuscript text omitted]
| Highly labile carbon in AL | [$GtCO_2$] $GtC$ | 45.3 34.1 | 48.0 37.5 | 52.1 42.7 | 54.1 45.9 | 48.0 40.2 | 44.9 36.0 | 42.4 36.1 | 39.4 35.3 | 36.7 35.5 |
| Labile carbon in AL | [$GtCO_2$] $GtC$ | 149.5 87.0 | 163.7 97.8 | 183.8 113.9 | 200.3 128.5 | 190.8 129.0 | 152.6 94.1 | 137.8 91.3 | 117.1 88.7 | 100.4 87.9 |
| Stable carbon in AL | [$GtCO_2$] $GtC$ | 303.2 124.1 | 349.0 148.6 | 428.3 194.5 | 523.8 251.6 | 570.5 286.2 | 332.7 137.1 | 339.7 136.4 | 337.4 136.3 | 326.2 131.1 |
| Net carbon flux | [$GtCO_2$] $GtC$ $year^{-1}$ | 0.14 -0.34 | 0.4 -0.32 | 1.0 -0.15 | 2.0 -1.03 | 2.3 -1.11 | 0.19 -0.36 | -0.02 -0.47 | -0.13 -0.47 | -0.14 -0.44 |
| Net primary productivity | [$GtCO_2$] $GtC$ $year^{-1}$ | 3.6 -3.91 | 4.3 -4.58 | 5.2 -5.59 | 6.6 -7.17 | 7.9 -8.7 | 3.9 -4.15 | 4.0 -4.33 | 4.2 -4.5 | 4.2 -4.59 |
| Soil CO$_2$ flux | [$GtCO_2$] $GtC$ $year^{-1}$ | 3.6 -4.39 | 4.4 -4.02 | 5.9 -5.43 | 8.2 -7.75 | 9.8 -9.2 | 3.8 -3.59 | 3.8 -3.67 | 3.8 -3.83 | 3.9 -3.95 |
| Soil CH$_4$ flux | [$MtCO_2$] $MtC$ $year^{-1}$ | 9.6 -6.9 | 9.9 -8.2 | 11.0 -11.1 | 11.2 -13.4 | 6.7 -6.0 | 8.1 -6.6 | 6.2 -4.7 | 4.0 -2.7 | 2.4 -1.28 |
| Soil methane production | [$MtCO_2$] $MtC$ $year^{-1}$ | 23.5 -23.0 | 27.4 -27.1 | 34.3 -35.5 | 42.1 -43.7 | 42.2 -39.3 | 23.2 -23.3 | 21.0 -20.9 | 18.0 -19.1 | 15.9 -17.7 |
| CH$_4$ plant transport | [$MtCO_2$] $MtC$ $year^{-1}$ | 10.9 -9.4 | 12.4 -11.2 | 15.0 -15.1 | 16.9 -18.2 | 13.5 -11.6 | 10.3 -9.4 | 8.6 -7.8 | 6.6 -6.0 | 5.0 -4.69 |
| CH$_4$ oxidation | [$MtCO_2$] $MtC$ $year^{-1}$ | 14.5 -16.1 | 17.5 -18.9 | 23.2 -24.4 | 30.9 -30.4 | 35.6 -33.3 | 15.4 -16.7 | 14.7 -16.2 | 14.0 -16.3 | 13.6 -16.4 |

**Table 2.** Continuation of table 1

[revised manuscript text omitted]

---

## Referee Report (RR1)

**A peer review to manuscript**
**Diverging responses of high latitude $CO_2$ and $CH_4$ emissions in idealized climate change scenarios**

The scientific rigour of the paper is improved upon revision. However, I guess that some technical details in the manuscript are excessive (see below). All my comments are minor:

- - l. 248: The notation '$z \neq s$' is difficult to read. Is it 'not at the surface'?

- ll. 315 and 326: it would be better 'standard deviation of subgrid-scale orography'

- I guess that ll. 414-439 are too techical. What it adds to the scientific results of the paper.

- ll. 437 and 438: if the part of the text indicated in the previous item is still retained of the text, it is better to replace subscript 'pysics' to 'physics'.

- l. 819: insert break between 'CO2' and 'concentration'.

- ll. 1116 and 1119: What is the difference between (Zimov, 2006a) and (Zimov, 2006b)?

---

## Author Response (AR3)

Hamburg, 27.01.2021

Dear Editor,

for the upload of the correction file we modified the manuscript according to the reviewer's comments. These were mainly of a more technical nature:

l. 248: The notation 'z $\neq$ s' is difficult to read. Is it 'not at thesurface'? We changed the notation to $X_{surf}$ for all variables at the surface allowing simply use $X_z$ for all below-ground variables at depth $z$.

ll. 315 and 326: it would be better 'standard deviation of subgrid-scaleorography'. Here, we changed the formulation to 'standard deviation of subgrid-scaleorography'.

l. 819: insert break between 'CO2' and 'concentration. The text was changed accordingly.

ll. 1116 and 1119: What is the difference between (Zimov, 2006a) and(Zimov, 2006b)? There, was a mix-up in our reference file which we since have corrected.

However, there was also one suggestion that lead a slightly larger modification of the manuscript. In the method section we described the parameter-setups that were used for the different simulations of our 20-member-ensemble. Here, the reviewer commented: I guess that ll. 414-439 are too techical. What it adds to the scientific results of the paper. (ll. 437 and 438: if the part of the text indicated in the previous item is still retained of the text, it is better to replace subscript 'pysics' to'physics'.) Here we fully agree with the reviewer, that the passage is very technical and may not be of interest to all the readers. However, we think that this is not only the case for the paragraph mentioned by the reviewer, but for the entire description of the ensemble. This aspect is only relevant for those readers that are interested in the uncertainty and how certain formulations affect the simulated dynamics. Thus, we moved the entire description of the ensemble to the Appendix, where also the respective uncertainty is discussed (note that we also corrected the typo 'pysics' to 'physics').

Finally, we started uploading our data to the long-term archive of the German Climate Computing Center as soon as we learned that the manuscript was excepted. However, we still have not obtained the permanent URL for the data, which we will hand in as soon as we receive it.

Again, thank you for your help with our manuscript.

With best regards,
Philipp de Vrese